# Critical role of hydrogen sorption kinetics in electrocatalytic CO$_2$ reduction revealed by on-chip in situ transport investigations

Zhangyan Mu[1,6], Na Han[2,3,6], Dan Xu[4], Bailin Tian[1], Fangyuan Wang[1], Yiqi Wang[1], Yamei Sun[1], Cheng Liu[1], Panke Zhang [5], Xuejun Wu [4], Yanguang Li [2,3] & Mengning Ding [1]

Precise understanding of interfacial metal−hydrogen interactions, especially under in operando conditions, is crucial to advancing the application of metal catalysts in clean energy technologies. To this end, while Pd-based catalysts are widely utilized for electrochemical hydrogen production and hydrogenation, the interaction of Pd with hydrogen during active electrochemical processes is complex, distinct from most other metals, and yet to be clarified. In this report, the hydrogen surface adsorption and sub-surface absorption (phase transition) features of Pd and its alloy nanocatalysts are identified and quantified under operando electrocatalytic conditions via on-chip electrical transport measurements, and the competitive relationship between electrochemical carbon dioxide reduction (CO$_2$RR) and hydrogen sorption kinetics is investigated. Systematic dynamic and steady-state evaluations reveal the key impacts of local electrolyte environment (such as proton donors with different p$K_a$) on the hydrogen sorption kinetics during CO$_2$RR, which offer additional insights into the electrochemical interfaces and optimization of the catalytic systems.

Metal−hydrogen (M−H) interactions and the correlated chemical/catalytic hydrogen processes (H adsorption, absorption, evolution, and oxidation) participate in multiple applications such as hydrogen fuel cells[1,2], hydrogen/pH sensors[3–6], metal hydride batteries[7], and electrocatalytic hydrogen evolution reaction (HER) and hydrogenation reactions[8–10]. To this end, Palladium (Pd) is one of the mostly adapted materials, which serves as a typical model catalyst for the fundamental investigation of M−H states and dynamic hydrogen transitions, owing to its unique and rich interactions with hydrogen[9–13]. Among various Pd-catalyzed electrochemical reactions, electrochemical carbon dioxide reduction (CO$_2$RR) attracts most research attentions as it represents a sustainable means to reduce CO$_2$ emissions by converting it into valuable chemicals and hydrocarbon fuels, providing an effective and economical approach towards carbon neutralization[14–17]. Numerous studies have been focused on the compositional and morphological innovations on Pd-based nanostructures[18–24] to obtain high current densities and Faradaic efficiencies (FEs) of desired products, and one particular effective approach is the use of bimetallic catalysts where alloying elements such as Ag[20,21] and Au[25,26] can alter the electronic structure of Pd, regulate the intermediate adsorption energy, and finally improve the CO$_2$RR performance. In analogy, H can be viewed as another alloying source for Pd: the formation of Pd−H bond involves a charge transfer process[16,18,22,24], followed by the consequent transition to PdH$_x$ as a separate phase. H atoms either adsorb on the surface or diffuse into the subsurface, and significantly alter the adsorption energy of reaction intermediates, such as *CO, *HCOO, and *COOH[18,24].

[1]Key Laboratory of Mesoscopic Chemistry, School of Chemistry and Chemical Engineering, Nanjing University, Nanjing 210023, P. R. China. [2]Institute of Functional Nano & Soft Materials (FUNSOM), Soochow University, Suzhou 215123, P. R. China. [3]Jiangsu Key Laboratory for Advanced Negative Carbon Technologies, Suzhou, China. [4]State Key Laboratory of Coordination Chemistry, School of Chemistry and Chemical Engineering, Nanjing University, Nanjing 210023, P. R. China. [5]State Key Laboratory of Analytical Chemistry for Life Science, School of Chemistry and Chemical Engineering, Nanjing University, Nanjing 210023, P. R. China. [6]These authors contributed equally: Zhangyan Mu, Na Han. ✉e-mail: yanguang@suda.edu.cn; mding@nju.edu.cn

Despite the significant influence on the adsorption of intermediates in CO$_2$RR and other hydrogenation reactions, there are only few experimental approaches for the quantitative measurement of adsorbed/absorbed H atoms and corresponding H sorption kinetics in Pd-based catalysts under operando conditions. Specifically, the phase transition of Pd is buried at a solid/liquid interface, which is difficult for in situ characterizations and poses a particular challenge in the study of corresponding electrocatalytic mechanisms. In most cases, in situ X-ray absorption spectroscopy (XAS) and in situ X-ray diffraction (XRD) were typically employed to characterize the Pd−Pd bond lengths and the lattice expansion during phase transition[17,18,22,27,28], which successfully revealed the impact of catalyst morphologies on the potential range (with difference up to 100-300 mV) for PdH$_x$ formation. Alternating current (AC) impedance[29], quartz crystal microbalance[30], and cyclic voltammetry[31–33] are the commonly employed approaches for directly studying Pd−H interactions, however, each individual methodology typically produces information on restricted dimension. To fully elucidate the comprehensive electrocatalytic mechanisms that include interfacial chemical processes and the local environments, it is essential to bring up additional in situ approaches (better with alternative signaling mechanism) to complement the existing characterization toolbox for the systematic investigation of Pd−H interactions and corresponding hydrogenation processes.

The transition between Pd and Pd hydride (α-phase or β-phase, PdH$_x$) occurs naturally in hydrogen atmosphere, which sharply increases the resistivity that is proportional to the H content. This serves as the basis for the fabrication of Pd-based H$_2$ gas sensors[4,6,34]. In principle, similar resistivity change would occur during electrochemical H sorptions in Pd-based nanomaterials. To this end, a recently developed in situ transport-based characterization technique, electrical transport spectroscopy (ETS), coupled with on-chip cyclic voltammetry (CV), enables the in situ electrical transport measurement of electrochemical interfaces, which is particularly suitable for the in situ investigation of H sorptions during electrocatalytic processes[35,36]. Here we report a comprehensive mechanistic study with quantitative measurements of interfacial hydrogen sorption processes in Pd-based CO$_2$RR, including both intermediate states of surface hydrogen adsorption and subsurface PdH$_x$ formation (i.e., diffusion and absorption of H into the Pd lattice), using in situ transport-based measurements on a microelectrochemical platform. By dynamically probing the in situ conductivities of Pd nanocatalysts in varying reactions and electrolytes, accompanied with rigorous electrochemical and electrokinetic investigations, we revealed the competitive relationship between CO$_2$RR and surface/sub-surface H processes of Pd-based catalysts, and elucidated the key impacting factor of electrolytes (proton donors with different p$K_a$) that determined the hydrogen sorption kinetics and CO$_2$RR performances. The new mechanistic understandings were further demonstrated to provide valuable insights into the principles of performance enhancement in alloying catalysts.

## Results and discussion
### Catalyst preparation and device fabrication

Pd (with other metals as comparison) and Pd$_4$Ag alloy were studied as model catalysts for the elucidation of H adsorption/absorption and catalytic activities. Pt, Pd, and Pd$_4$Ag nanowires were prepared according to the previously reported methods[20,37,38] with slight modifications (see Methods). The transmission electron microscopy (TEM) images of the as-synthesized catalysts are shown in Fig. 1c. The diameters of Pt, Pd, and Pd$_4$Ag nanowires are 3 nm, 8 nm, and 6 nm, respectively. The crystalline structures were studied by X-ray diffraction (XRD). As shown in Fig. S1, the diffraction peaks of pure Pd are well consistent with (111), (200), (220), (311), and (222) planes of Pd fcc crystal structure. The diffraction peaks of Pd$_4$Ag are located between pure Pd and Ag, but are closer to Pd, indicating the formation of Pd-rich alloy[20,39]. Inductively coupled plasma (ICP) analysis suggests that Pd$_4$Ag alloy is composed of 80.96 at.% of Pd and 19.04 at.% of Ag, consistent with the starting Pd/Ag molar ratio. The interaction between Pd and Ag in Pd$_4$Ag was further probed by X-ray photoelectron spectroscopy (XPS). The Pd 3$d$ doublet of Pd$_4$Ag shifts to lower binding energy as compared to pure Pd, indicating the electron

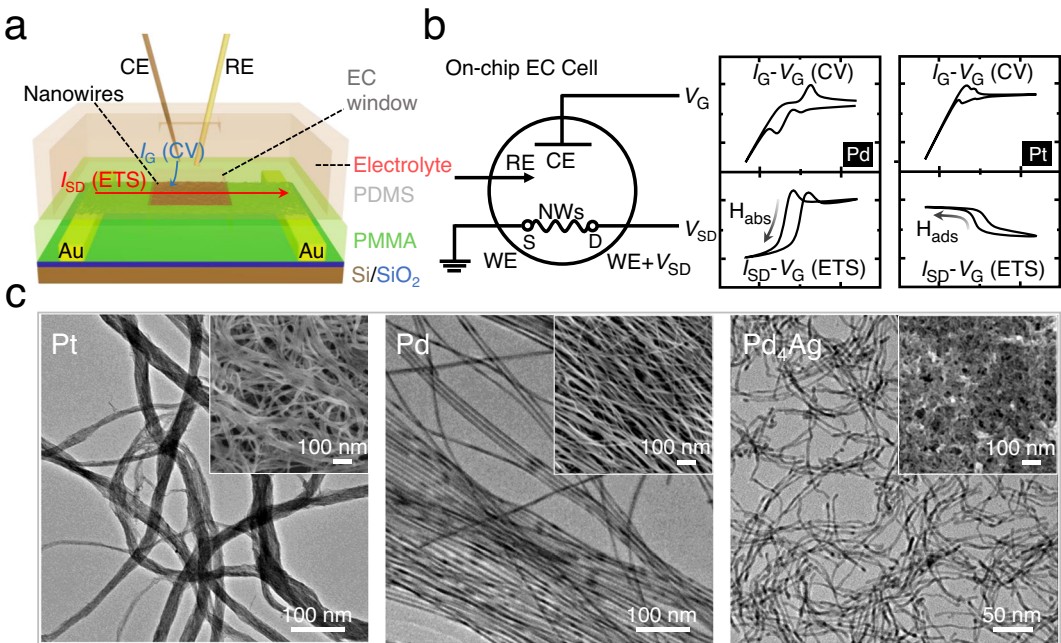

**Fig. 1 | Working principle of on-chip electrochemical and in situ electrical transport measurements during H adsorption and absorption (subsurface PdH$_x$ formation). a** Schematic illustration of the structure of the integrated electronic device and on-chip electrochemical (EC) cell. **b** Schematic illustration of circuits (left) and representative results (right) in concurrent CV and ETS measurements of Pt and Pd-based nanocatalysts. H$_{ads}$ and H$_{abs}$ represent the adsorbed H on the surface and absorbed H in the bulk, respectively. **c** Typical TEM images of the as-synthesized Pt, Pd, and Pd$_4$Ag nanowires. The insets show SEM images of the on-chip nanowire films. CE, counter electrode; RE, reference electrode; WE, working electrode; S, source; D, drain.

transfer from Ag to Pd, resulting in negatively charged $Pd^{\delta-}$ (Fig. S2), which can be rationalized by the lower work function and higher electron density of Ag[20].

Figure 1a, b depicts the schematic experimental setup and working principle for the concurrent on-chip CV and ETS measurements. A two-channel source-measure-unit (SMU, Key-sight 2902a) was employed for ETS measurements, with simultaneous $I_G - V_G$ (CV) and $I_{SD} - V_G$ (ETS) readouts (Fig. S3). More experimental details can be found in Methods. The devices were fabricated by selectively depositing the nanowires films onto the Si wafer with prepatterned gold electrodes[35,40] (Fig. S4). For in situ electrical transport measurements, the as-prepared devices were then covered with an inert layer (PMMA) to define an electrochemical window (Fig. S4). Typical optical microscopic (OM) images of on-chip nanowires are shown in Fig. S5. For the stable measurements of hydrogen phenomena especially during electrocatalytic processes, the mean thicknesses of the films were typically controlled at about 200 nm (Fig. S5).

## ETS identification of in situ H sorption processes in perchloric acid

Surface hydrogen adsorption and subsurface hydride formation were first investigated in 0.1 M $HClO_4$ (Fig. 2). Stable CV and ETS curves of Pt, Pd and $Pd_4Ag$ were acquired after several cycles of electrochemical activation (Fig. S6). As shown in Fig. 2a, CV and ETS of Pt show characteristic behaviors similar to the previously reported results on polycrystalline Pt surface[35,36,40], serving as a convenient baseline for the further investigation of other unknown systems. The ETS curve of Pt can be typically divided into three regions: H adsorption and evolution region (region I), double-layer region (D.L., region II), and reversible adsorption of hydroxyl groups and surface oxide formation region (M−OH and M−O, region III). For the ease of analysis, the ETS curves Pd and $Pd_4Ag$ are correspondingly divided into these same regions (Fig. 2b,

c). Notably, significant differences in region I can be observed for the three materials, indicating the diverse phenomena of M−H interactions. With the gradually reducing potential, the ETS current ($I_{SD}$) of Pt rises first and then reaches to a plateau (pink arrow in Fig. 2a), a typical indication of a stabilized state of saturated monolayer surface hydrogen adsorption (Pt−H)[35]. In sharp contrast, the ETS current of Pd rises first but then shows a unique and dramatic decline to a plateau at a much lower level (green arrow in Fig. 2b). As previously established in the studies of Pt[35,36,40], when the potential decreased from 0.5 $V_{RHE}$ (D.L. region) to less than 0 $V_{RHE}$ (HER region), the adsorbate on the Pt surface changed from water to active hydrogen ($H_{ads}$). Due to the less diffusive scattering of charge carriers from a Pt−H surface than a Pt−$H_2O$ surface[35,36,40,41], higher $I_{SD}$ can be obtained (schematic illustration presented in Fig. 2e). In principle, this phenomenon can also be observed on other metals including Pd that form strong surface (covalent) M−H bonding. Therefore, at a less negative potential higher than 0.07 $V_{RHE}$, ETS of Pd mainly reflects the H adsorption on surface. However, when potential continues to decline, the H diffuses into the lattice and leads to the formation of hydride (phase transition) that is unique to the Pd system, resulting in additional electron scattering (Fig. 2e) and decrease in density of states (DOS) at the Fermi level[12,34], thus the decline of $I_{SD}$. Fortunately, the opposite conductivity trend from surface H adsorption and consequent hydride formation offer a clear, convenient and sensitive approach for the detailed in situ mechanistic investigation of H-involving processes in Pd-catalyzed electrochemical reactions.

Generally, the phase transition from Pd to $PdH_x$ can be quantified by the ratio of Pd and absorbed H atoms, which can be obtained through calculating the Faradaic quantity during H electro-oxidation on a well-defined electrode[31-33,42]. Lasia et al. separated H adsorption and absorption on Pd films with different thicknesses deposited on Au(111), and concluded that the $\beta$-$PdH_x$ is obtained at potentials lower than 0.05 $V_{RHE}$ while the $\alpha$-$PdH_x$ is obtained at more positive

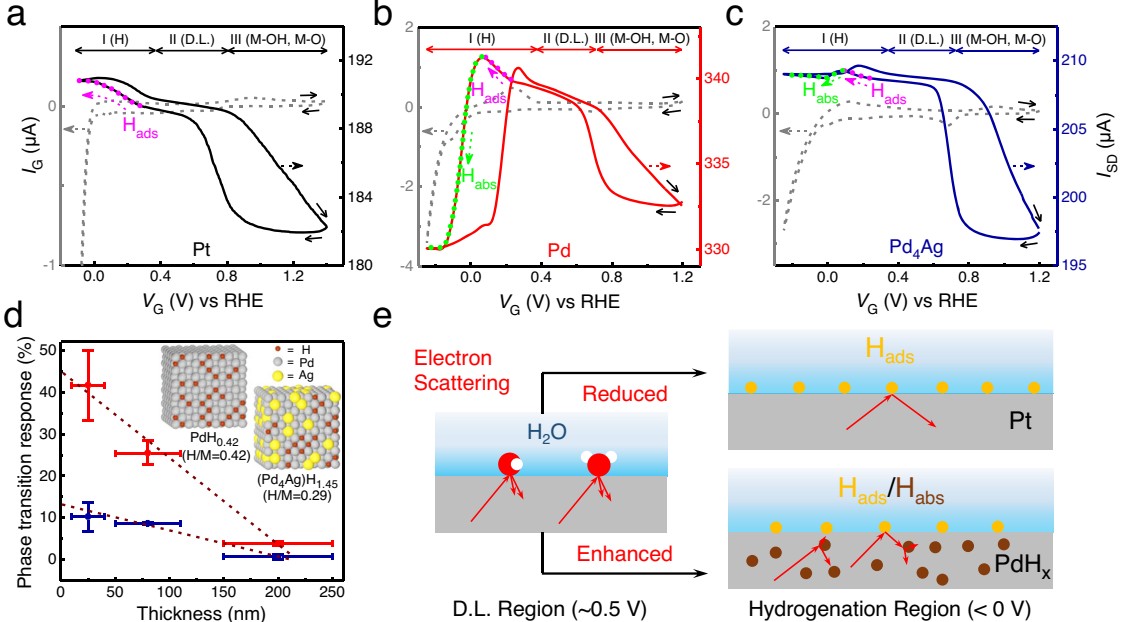

**Fig. 2 | Electrochemical interfacial H processes in acidic condition. a-c** $I_G$-$V_G$ (CV, dash) and $I_{SD}$-$V_G$ (ETS, solid) curves of Pt (**a**), Pd (**b**), and $Pd_4Ag$ (**c**) in 0.1 M $HClO_4$. Film thickness is ~200 nm. I, II, and III represent the different states of the metals. Solid arrows in (**a-c**) indicate the potential sweeping direction. **d** The relationship between thicknesses of Pd (red) and $Pd_4Ag$ (blue) nanowire films and phase transition responses ($\triangle R_{MHx}$) reflected on ETS. Inset shows the schematic illustration of hydride with H/M ratio of 0.42 in $PdH_x$ and 0.29 in $(Pd_4Ag)H_x$. Ag randomly occupies the Pd position and H preferentially occupies the octahedral

vacancy close to Pd. The error bars in (**d**) present the standard error in nanowire film thicknesses and ETS experiments. **e** Schematic illustration of electron scattering (red arrows) in metal with surface adsorbates and hydrides. The black arrows indicate the decrease or increase of electron scattering during the change of states of electrodes. The yellow and brown cycles represent the adsorbed H on the surface ($H_{ads}$) and absorbed H in the bulk ($H_{abs}$), respectively. The red and white cycles represent the oxygen and hydrogen atoms of water molecules, respectively. D.L. region, double layer region. Source data are provided as a Source data file.

potentials[31]. It is also found that the generation of α- and β-PdH$_x$ both leads to decreases of conductivity[12]. However, no obvious decline of $I_{SD}$ was observed at the relatively positive potentials (0.07–0.30 V$_{RHE}$) on Pd (Fig. 2b). This indicates the low level of absorbed H, and the rise of $I_{SD}$ caused by H$_{ads}$ is dominant in this potential range during the dynamic (non-equilibrium) CV scan. Meanwhile, the onset potential of 0.07 V$_{RHE}$ on ETS (highlighted by green color in Fig. 2b) corresponds to the rapid phase transition (PdH$_x$ formation), which is consistent with the reported result (0.05 V$_{RHE}$)[31], indicating the successful detection of H adsorption and consequent absorption by ETS.

Since the precise number of Pd atoms varies in each device, it is difficult to quantitatively describe PdH$_x$ simply by H electro-oxidation. We can quantify the phase transition level of Pd through its resistance change (phase transition response):

$$\Delta R_{MH_x} = \frac{R_{sat} - R_{onset}}{R_{onset}} 100\% \tag{1}$$

where $R_{sat}$ is the resistance of catalyst film fully saturated by absorbed H (H$_{abs}$), and $R_{onset}$ is the resistance of catalyst film at the onset potential for rapid phase transition. The $\triangle R_{PdHx}$ of pure Pd system within −0.20–0.07 V$_{RHE}$ is calculated to be 3.40% (Fig. 2b), which is obviously lower than the widely accepted value (>70%) for the full β-PdH$_x$ formation[12,43]. This is probably due to the insufficient electrolyte and hydrogen diffusion to the underlying layer of nanowires when the film thickness is large in the devices. To address this issue, film responses were systematically measured with the varying thicknesses of nanowires films to establish a calibration. We found that $\triangle R_{PdHx}$ was highly relevant to the film thickness with a linear correlation (Fig. 2d, Fig. S9). The intercept of 45% represents the maximum in situ phase transition response of Pd in 0.1 M HClO$_4$ electrolyte environment. Further increase in the size of electrochemical window does not significantly affect $\triangle R_{PdHx}$ (Fig. S10).

As for the Pd$_4$Ag alloying catalyst, the onset potential for H absorption is at 0.074 V$_{RHE}$, which is close to that of Pd. However, the $I_{SD}$ drop in response to hydride formation is considerably lower (Fig. 2c), and the $\triangle R_{(Pd4Ag)Hx}$ is calculated to be only 0.14%, which is about 4.12% of $\triangle R_{PdHx}$, clearly indicating a significantly different M–H interaction compared to pure Pd[44]. As shown in Fig. 2d, the $\triangle R_{(Pd4Ag)Hx}$ is lower than that of pure Pd at each film thickness, with a theoretical intercept of 13%. The M–H interactions in Pd$_4$Ag and pure Pd were further revealed by DFT calculations, as shown in Fig. S13 and Table S1. With a much weaker Ag–H interaction, the alloying Ag atoms can reduce the H adsorption and absorption in Pd$_4$Ag from both electronic structure and proximity effects. As an alloying element, Ag does not change the lattice distance of Pd$_4$Ag significantly, as evidenced by XRD results (Fig. S1). However, with the electron transfer between Ag and Pd in Pd$_4$Ag, the d orbital of Pd is filled with more electrons compared to pure Pd, weakening the ability of Pd to bond with H and reducing the resistivity change caused by Pd–H interactions[44]. Based on the on-chip $\triangle R_{MHx}$ of Pd and Pd$_4$Ag, we can estimate the number of the absorbed H atoms by referring to the known relationship between the resistivity and H/M ratio (H/M represents the ratio of hydrogen atoms to the combined total of Pd and Ag atoms) of Pd$_{80}$Ag$_{40}$ alloy and pure Pd (Fig. S14)[43,45]. The on-chip $\triangle R_{PdHx}$ of 45% corresponds to an H/M ratio of 0.42, and $\triangle R_{(Pd4Ag)Hx}$ of 13% corresponds to an H/M ratio of 0.29. The lower in situ H/M ratio of Pd$_4$Ag shows that the doping of Ag weakens the phase transition from Pd to Pd hydride, which provides solid experimental evidence to the weakened M–H interaction and thus phase transition.

## Electrochemical CO2RR performances and their kinetic dependence on H sorptions in buffered electrolytes

Based on the identification and quantification of in situ H sorption processes in the acidic electrolyte enabled by transport measurements, we next aim to study the H sorption impacts on Pd-catalyzed CO$_2$RR in a commonly used bicarbonate buffer electrolyte (KHCO$_3$), and a similar phosphate buffer electrolyte (K$_2$HPO$_4$/KH$_2$PO$_4$) for comparison. The CO$_2$RR product distributions were firstly obtained under constant potential electrolysis from −0.5 to 0.0 V$_{RHE}$, as shown in Fig. 3a–c. In KHCO$_3$, formate starts to form at ~0 V$_{RHE}$ on Pd with low FE of 35.87%, which increases to 97.54% at −0.33 V$_{RHE}$ and then drops back to 42.47% at −0.42 V$_{RHE}$. In comparison, formate also starts to form on Pd$_4$Ag at ~0 V$_{RHE}$ but with an obviously higher FE of 89.14%, and FE remains at high level (>80% between 0 to −0.30 V$_{RHE}$, 72.64% at −0.43 V$_{RHE}$) at the high overpotential region (indicating the resistance to competing HER). Moreover, the chronoamperometric (i–t) curves of Pd$_4$Ag indicates a high working stability, whereas the current density of pure Pd starts to decline with working potential less than −0.23 V$_{RHE}$ owing to the CO poisoning, in line with previous reports[15–17,20] (Fig. S15). Figure 3b further shows that Pd$_4$Ag has higher formate production rates than Pd at all tested potentials. The above results indicate an overall better CO$_2$RR performance after the alloying with Ag atoms.

Interestingly, when the electrolyte was switched to K$_2$HPO$_4$/KH$_2$PO$_4$, the overall formate FEs on pure Pd were reduced (Fig. 3a). In comparison, Pd$_4$Ag showed even more complex changes in CO$_2$RR performance. The formate FEs also experienced an obvious decrease at low overpotentials (−0.15 to ~0 V$_{RHE}$), yet a unique increase by ~8% was observed at the high overpotential region (−0.4 to −0.2 V$_{RHE}$). Overall, the different CO$_2$RR performances in KHCO$_3$ and K$_2$HPO$_4$/KH$_2$PO$_4$ indicate distinct interfacial processes sensitive to the proton-donating electrolytes, which are presumably connected to the phase transition level of catalysts, CO poisoning/site blocking, formate production rates (Fig. 3b) and HER kinetics (Fig. 3c). It should be noted that the XRD after chronoamperometric studies reveal that H sorption processes during CO$_2$RR alloy does not cause a segregation (or any other structural or compositional change) in the Pd$_4$Ag (Fig. S17), indicating its reversibility during the reaction.

ETS measurements were further conducted to elucidate the H sorption kinetics and rationalize the distinct CO$_2$RR performances of different Pd-based catalysts in different electrolyte environments. First, we compared the H sorption kinetics by ETS in Ar-saturated KHCO$_3$ and HClO$_4$ at varying scan rates (Figs. 3d and S18). In HClO$_4$, significantly larger ETS hysteresis loops were observed to achieve complete H adsorption/desorption when the scan rate was increased from 10 to 80 mV/s, which confirms that the H sorption process is kinetic-dependent. In addition, while the degree of phase transition (corresponding to the ETS current level) kept unchanged in HClO$_4$, a lower degree of phase transition (dashed arrow in Fig. 3d) was observed in KHCO$_3$ with the increasing scan rate. These results suggest the slow H sorption kinetics in the near neutral electrolyte, which is reasonable due to the low concentration of hydronium ion and slow kinetics of water reduction in neutral/alkaline electrolytes[46–49].

Figure 3e depicts the CV and ETS curves of Pd under CO$_2$RR conditions (in 0.1 M CO$_2$-saturated KHCO$_3$). An obvious alternation in the H sorption hysteresis loop in ETS can be observed (marked with blue colored area and dashed arrow in Fig. 3e) after the introduction of CO$_2$, which may originate from the change of proton source and competitive surface reactions. Specifically, the addition of CO$_2$ increases the concentration of H$_2$CO$_3$ in the electrolyte:

$$CO_2 + H_2O \rightleftharpoons H_2CO_3 \tag{2}$$

and the Faradaic current concerning H-containing species thus follows:

$$j_H = j_{H_2CO_3} + j_{HCO_3^-} + j_{H_2O} + j_{H_3O^+} \tag{3}$$

The contribution by hydronium ions can be ignored as its concentration is relatively low (≤10$^{-6}$ M) in near neutral electrolytes[50]. The introduction of CO$_2$ changes the pH of electrolyte from 8.3 to 6.8. A

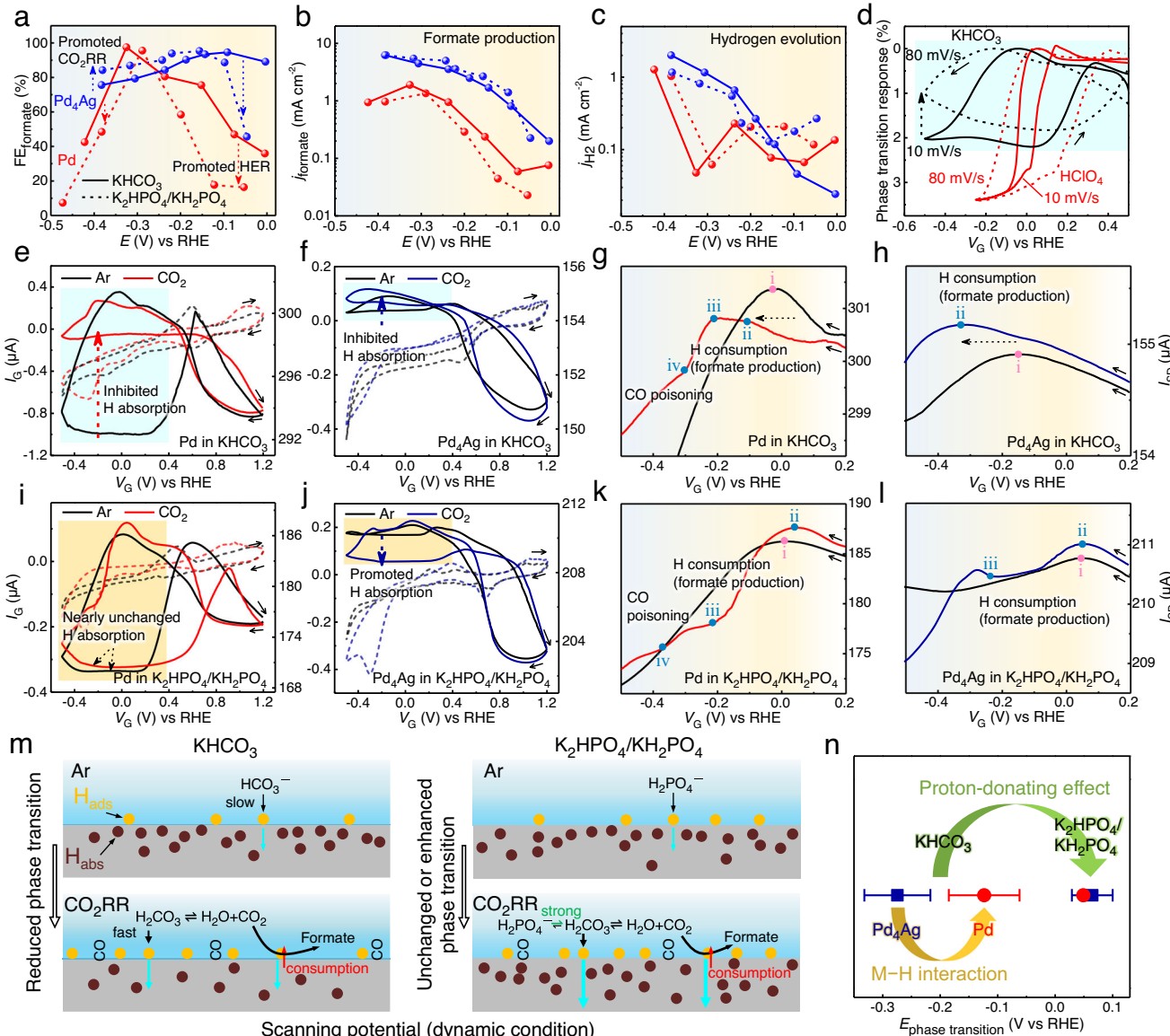

**Fig. 3 | CO₂RR and H sorption processes in KHCO₃ and K₂HPO₄/KH₂PO₄ electrolytes. a–c** Potential-dependent selectivity for formate production (**b**), and current densities for formate production (**b**), and hydrogen evolution (**c**) with Pd (red) and Pd₄Ag (blue) catalysts in 0.1 M CO₂-saturated KHCO₃ (solid curves) and K₂HPO₄/KH₂PO₄ (dash curves). **d** Phase transition responses of Pd in 0.1 M Ar-saturated KHCO₃ and HClO₄ with varying scan rates. Film thickness is ~200 nm. **e–h** On-chip $I_G$–$V_G$ (CV, dash, left y axis) and $I_{SD}$–$V_G$ (ETS, solid, right y axis) curves of Pd (**e, g**) and Pd₄Ag (**f, h**) in 0.1 M Ar- and CO₂-saturated KHCO₃. **g** and **h** depict the enlarged negative-potential-sweeping ETS curves (0.2 to −0.5 $V_{RHE}$) in (**e**) and (**f**), respectively. **i–l** On-chip $I_G$–$V_G$ (CV, dash, left y axis) and $I_{SD}$–$V_G$ (ETS, solid, right y axis) curves of Pd (**i, k**) and Pd₄Ag (**j, l**) in 0.1 M Ar- and CO₂-saturated K₂HPO₄/KH₂PO₄. **k** and **l** depict the enlarged negative-potential-sweeping ETS curves (0.2 to −0.5 $V_{RHE}$) in (**i**) and (**j**), respectively. Solid arrows in (**d–l**) indicate the potential sweeping direction. Film thickness is ~200 nm. The gradient background (from red to blue color) in (**a–c, g, h, k, l**) represents the increased CO poisoning effect concluded from Figs. S15, 16. The rectangular shadings (in **d, e, f, i, j**) highlight the H sorption regions reflected on ETS in specific electrolytes (green in KHCO₃ and red in K₂HPO₄/KH₂PO₄). **m** Schematic illustration of different Pd–H states in KHCO₃ (left panel) and K₂HPO₄/KH₂PO₄ (right panel), and the corresponding CO₂RR processes at the interfaces. The yellow and brown cycles represent the adsorbed H on the surface (H_{ads}) and absorbed H in the bulk (H_{abs}), respectively. **n** Summary of phase transition potentials of Pd (blue) and Pd₄Ag (red) under CO₂RR conditions in KHCO₃ and K₂HPO₄/KH₂PO₄ obtained at 10 mV/s. The error bars in (**n**) present the standard error in two ETS experiments with film thicknesses of ~25 nm and ~200 nm. Source data are provided as a Source data file.

neutral condition is not beneficial to the hydrogen kinetics due to the low hydronium ion concentration and the insufficient driving force for H₂O reduction[49], which further contributes to the negatively shifted potential for hydrogen adsorption (dashed arrow in Fig. 3g) and phase transition potential (i and ii in Fig. 3g) reflected on ETS. Moreover, H₂CO₃ has a lower p$K_a$ than HCO₃⁻ and H₂O (Table 1), which allows it to act as the first proton donor (PD) for H sorptions, as the p$K_a$ of a PD is related to the thermodynamic driving force for proton-donating[50]. This effect tends to shift the potential of H adsorption to a more positive position, which is, however, contrary to our ETS results

(dashed arrow in Fig. 3g). Furthermore, the onset potential for phase transition splits from point i (Fig. 3g) to two points (ii and iii in Fig. 3g) in CO₂-saturated KHCO₃, and the same phenomenon can be observed in successive cycles as shown in Fig. S19. It is only when the negative potential goes beyond point iii (Fig. 3g) that strong H absorption occurs. The non-linear variation of $I_{SD}$ and phase transition with the negative shift of potential therefore strongly indicate the competition between CO₂RR and H sorption at potentials <0 $V_{RHE}$ in KHCO₃. Two parallel pathways were generally proposed for Pd-catalyzed CO₂RR, leading to formate or CO products[24]:

**Table 1 | p$K_a$ of proton donors in the electrolytes**

|        | $H_3PO_4$ | $H_2PO_4^-$ | $HPO_4^{2-}$ | $H_2CO_3$ | $HCO_3^-$ | $H_2O$ |
|--------|-----------|-------------|--------------|-----------|-----------|--------|
| p$K_a$ | 2.15      | 7.21        | 12.35        | 6.35      | 10.33     | 14     |

in the formate or formic acid pathway:

$$CO_2 + e^- + H^+ + {}^* \rightarrow HCOO^* \tag{4}$$

$$HCOO^* + e^- + H^+ \rightarrow HCOOH^* \tag{5}$$

$$HCOOH^* \rightarrow HCOOH(aq) + {}^* \tag{6}$$

in the CO pathway:

$$CO_2 + e^- + H^+ + {}^* \rightarrow {}^*COOH \tag{7}$$

$${}^*COOH + e^- + H^+ \rightarrow {}^*CO + H_2O(l) \tag{8}$$

$${}^*CO \rightarrow CO + {}^* \tag{9}$$

The formate mechanism involves a proton-coupled electron transfer (PCET) process, during which the proton transfer and electron transfer occur in a same elementary step, and the M–H bond is formed on the surface[51,52]. Although H adsorption on the surface is thermodynamically more favorable (by 0.33 eV) compared to subsurface H absorption (which leads to hydride formation), H could diffuse into the subsurface and then bulk fcc Pd lattice at more negative potentials[18,53,54]. The overall hydrogen sorption process within Pd and Pd$_4$Ag systems can be described by[55]:

$$H^+ + M + e^- \rightleftharpoons MH_{ads} \rightleftharpoons MH_{subsurface}(MH_x) \tag{10}$$

where the diffusion between $H_{ads}$ and $H_{subsurface}$ follows the equilibrium that is determined by the chemical potentials of H atoms in each phase ($\mu_{Hads}$ vs. $\mu_{Hsubsurface}$). During active CO$_2$RR, the C1 intermediates occupy the Pd sites and inhibit the production of $H_{ads}$, shift the equilibrium between $H_{ads}$ and $H_{subsurface}$, and eventually alter the level of hydride formation. In addition, the more favorable formate pathway (Eqs. 4–6) will largely consume $H_{ads}$ and slow down the kinetics of subsurface H diffusion and phase transition process. Importantly, these H-involving processes during CO$_2$RR can be reflected on the ETS signals corresponding to the H sorptions. On this basis, as the unusual change in $I_{SD}$ (Fig. 3g) is in well correspondence with the high formate FE (>70%) in the potential range (>−0.2 V$_{RHE}$), our results therefore confirm that proton consumption and site blocking by intermediate adsorptions during formate production significantly reduce the H diffusion kinetics and level of phase transition under scanning potential (non-equilibrium) condition. When the potential continues to decrease to a more negative potential (−0.3 V$_{RHE}$, indicated by iv in Fig. 3g), the decline of $I_{SD}$ is slowed down (leading to a clear two-stage, non-linear ETS characteristic within the range of −0.2 to −0.4 V$_{RHE}$) probably due to severe CO poison, which inhibits the production of $H_{ads}$ and subsequent H absorption[32]. Overall, the near neutral electrolyte condition in KHCO$_3$ and the CO$_2$RR process both reduce the kinetics of H sorption processes under scanning potentials (as illustrated in Fig. 3m), finally causing a smaller ETS hysteresis loop (Fig. 3e). In such case, the excess of proton-donating KHCO$_3$ could facilitate the H sorption kinetics and enlarge the loop (Fig. S20). Similar competition also exists in Pd-catalyzed hydrogenation of formaldehyde, benzaldehydeand, benzonitrile, etc.[56,57].

Similarly, Fig. 3f depicts the CV and ETS curves of Pd$_4$Ag in 0.1 M Ar- and CO$_2$-saturated KHCO$_3$ that reveal the H sorption kinetics during

CO$_2$RR. The onset potential for phase transition shifted negatively from −0.145 V$_{RHE}$ (i in Fig. 3h) to −0.324 V$_{RHE}$ (ii in Fig. 3h) after the introduction of CO$_2$ in the electrolyte, and both are considerably lower than that of pure Pd. The more negative phase transition potentials of Pd$_4$Ag further confirm its weakened M−H interaction after Ag alloying. In addition, no change of fine ETS characteristics at high overpotentials (−0.2 to −0.4 V$_{RHE}$) was observed, which indicates unobvious CO poisoning effect, and H sorption kinetics is expectedly hindered due to the H consumption for intense formate production (dashed arrow in Fig. 3h). These in situ observations are also consistent with the strong CO poisoning resistance in Pd$_4$Ag alloying catalysts, which results in the considerably improved conversion rate and stability for CO$_2$RR (Figs. S15, 16). As shown in Fig. S13 and Table S1, our DFT calculation results also confirm that Ag can reduce the binding energy of poisonous *CO at its surrounding sites[20,21,25]. Finally, for similar reason to the Pd case, the inhibition by CO$_2$RR leads to a smaller ETS hysteresis loop for H sorption in Pd$_4$Ag (indicated by dashed arrow in Fig. 3f).

ETS investigations were further conducted in K$_2$HPO$_4$/KH$_2$PO$_4$ electrolyte to better clarify the fundamental connections between CO$_2$RR and H sorption processes. As shown in Fig. 3i, the similar hysteresis loops (marked with red area) suggest that the H absorption in pure Pd is not inhibited by CO$_2$RR in K$_2$HPO$_4$/KH$_2$PO$_4$ environment, which is in sharp contrast to the KHCO$_3$ case. Correspondingly, H absorption in Pd$_4$Ag is even accelerated by the introduction of CO$_2$ (dashed arrow in Fig. 3j). These results can be rationalized by the one major variation in these electrolytes, i.e., different proton-donating capacities of the corresponding anions. As shown in Table 1, anions with smaller p$K_a$ have stronger proton-donating capacity and tend to accelerate the H sorption kinetics. In Ar-saturated electrolytes, the onset potential for phase transition of Pd in K$_2$HPO$_4$/KH$_2$PO$_4$ (0.012 V$_{RHE}$, i in Fig. 3k) is higher than that in KHCO$_3$ (−0.024 V$_{RHE}$, i in Fig. 3g), which is consistent with the stronger proton-donating capacity of H$_2$PO$_4^-$ over HCO$_3^-$ without the influence of CO$_2$RR inhibition. For Pd$_4$Ag, the onset potential for phase transition (without CO$_2$RR) is also largely shifted from −0.145 V$_{RHE}$ in KHCO$_3$ (i in Fig. 3h) to 0.052 V$_{RHE}$ in K$_2$HPO$_4$/KH$_2$PO$_4$ (i in Fig. 3l). Under CO$_2$RR conditions, the addition of CO$_2$ can increase the local concentration of H$_2$CO$_3$ which has low p$K_a$ (6.35) and strong proton-donating capacity for H sorptions. In this case, H$_2$PO$_4^-$ (with lower p$K_a$ of 7.21) tends to more efficiently maintain the concentration of interfacial H$_2$CO$_3$ through equilibrium[50]:

$$HCO_3^- + H_2PO_4^- \rightleftharpoons H_2CO_3 + HPO_4^{2-} \tag{11}$$

thus supporting the H sorption processes and causing even stronger H absorptions during CO$_2$RR (dashed arrow in Fig. 3j). For the same reason, the onset potentials for phase transition in Pd and Pd$_4$Ag both show a slight positive shift during CO$_2$RR (0.012 to 0.044 V$_{RHE}$, 0.052 to 0.056 V$_{RHE}$, respectively), as shown in Fig. 3k, l (i and ii). In addition to the proton-donating effect, the influence of formate production and CO poisoning (as a result of the CO$_2$RR process) on H sorption in K$_2$HPO$_4$/KH$_2$PO$_4$ is also presented on ETS curves of Pd (ii, iii and iv in Fig. 3k, respectively). Again for Pd$_4$Ag, no obvious CO poisoning signal (lake of point iv in Fig. 3l) is reflected on ETS in K$_2$HPO$_4$/KH$_2$PO$_4$.

As summarized in Fig. 3m, while the H sorption kinetics is inhibited by CO$_2$RR-related H consumption and/or CO poisoning, it can be promoted by the local proton-donating species including H$_2$CO$_3$ in equilibrium with CO$_2$. It should also be noted that the H sorption is essentially a kinetic-dependent process, and the phase transition potentials may vary under different test conditions (potential scan rates, electrode geometries, etc.). Additional ETS tests indeed show positive shifts of the phase transition potential at smaller film thickness or slower scan rates (Fig. S21), emphasizing the importance of consistence in test conditions. To this end, the ETS measurements conducted with 25 nm thin film thickness and 10 mV/s scan rate represent

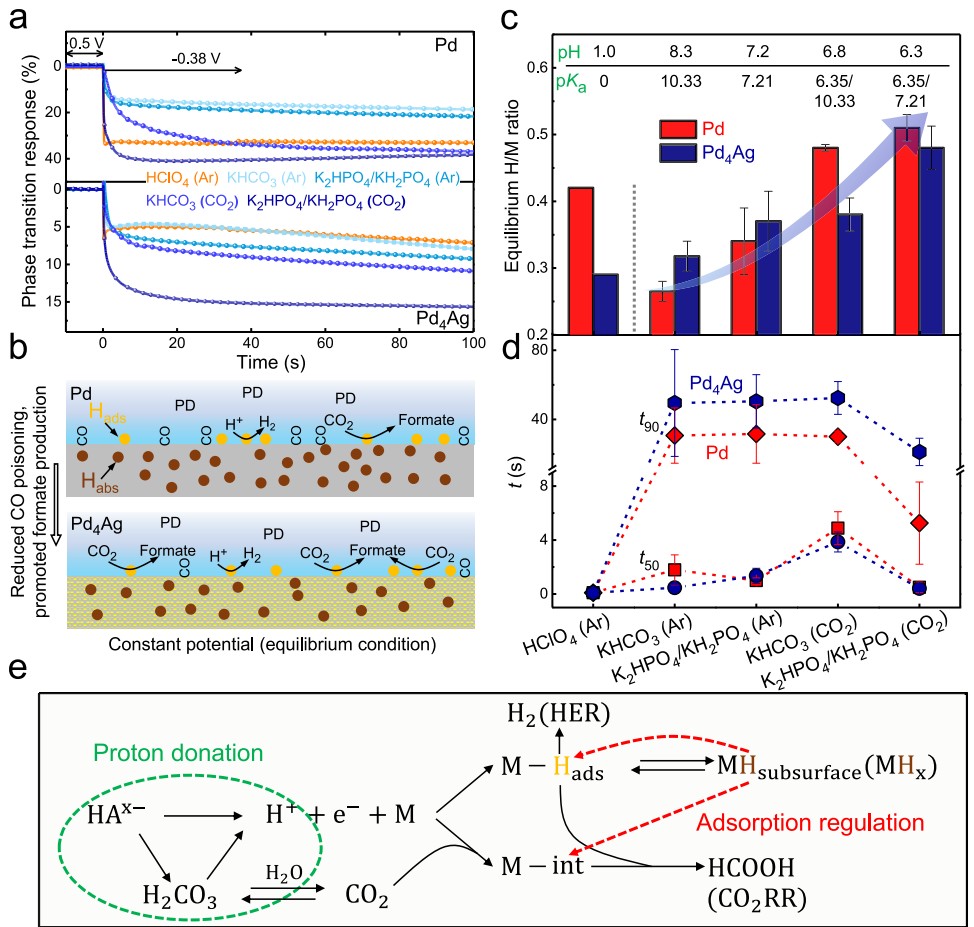

**Fig. 4 | Transient ETS quantifications of H sorption processes and CO₂RR per-formances during potentiostatic electrolysis. a** Phase transition response of Pd and Pd₄Ag under potentiostatic conditions in different electrolytes (0.1 M). The potential was first kept at 0.5 $V_{RHE}$ for 10 s and then shifted to −0.38 $V_{RHE}$ for 100 s. Film thickness is -25 nm. **b** Schematic illustration of M−H states and CO₂RR per-formances of Pd and Pd₄Ag in near neutral conditions. "PD" in (**b**) represents proton donor. The yellow and brown cycles represent the adsorbed H on the sur-face ($H_{ads}$) and absorbed H in the bulk ($H_{abs}$), respectively. **c** Summary of equili-brium H/M ratios obtained in different electrolytes at 100 s. The H/M ratios are obtained by first normalizing the responses in (**a**) with reference to the maximum phase transition responses (Pd: 45%, Pd₄Ag: 13%) obtained in 0.1 M HClO₄, and then

referring to the published quantitative relationship between the resistivity and H/M ratios of pure Pd and Pd₈₀Ag₂₀ alloy (Fig. S14). On the top shows the pH and p$K_a$ values of the proton donors with relatively high proton-donating capacities. The p$K_a$ of 0, 10.33, 7.21, and 6.35 corresponds to $H_3O^+$, $HCO_3^-$, $H_2PO_4^-$, and $H_2CO_3$, respectively. **d** Summary of the time for 50% and 90% level of maximum phase transition in Pd and Pd₄Ag. The error bars in (**c**, **d**) present the standard errors in two repeated ETS experiments. **e** Proposed mechanism for proton donation, H sorption and CO₂RR on Pd-based materials in near neutral conditions. "M−int" in (**e**) represents the surface adsorbed intermediates on metals during CO₂RR. Source data are provided as a Source data file.

---

the experimental condition that gives close-to-intrinsic properties of the electrode materials, where the impact from the insufficient elec-trolyte diffusion to the sub-layer nanowires within the thin film device was minimized. With precise control of these experimental factors, the phase transition potential of Pd was determined to be about 150 mV more positive than that of Pd₄Ag in KHCO₃, reflecting the different M−H interactions. In addition, by switching the electrolyte from KHCO₃ to K₂HPO₄/KH₂PO₄, the phase transition potentials of both Pd and Pd₄Ag markedly shift to 0–0.1 $V_{RHE}$, reflecting the strong proton-donation and fast H sorption kinetics in K₂HPO₄/KH₂PO₄. The key results and the corresponding conclusions were summarized in Fig. 3n.

**Potentiostatic ETS analysis for near-equilibrium operando H sorption quantifications**
The above ETS investigations in KHCO₃ and K₂HPO₄/KH₂PO₄ were all conducted with scanning potentials (CV), revealing the in situ com-petitive relationship between CO₂RR and H sorption processes under dynamic, diffusion-controlled, and non-equilibrium conditions. To further analyse and quantify the level of operando subsurface H absorption in Pd-based materials at their equilibrium conditions

during CO₂RR, we carried out transient ETS measurements during potentiostatic electrolysis, and the results are shown in Fig. 4a. The potential was first kept at 0.5 $V_{RHE}$, where the ETS signal under no phase transition serves as baseline, and was then switched to −0.38 $V_{RHE}$, which represents the typical potential for formate production with relatively high current density and selectivity. In all electrolytes, a clear drop in ETS currents can be observed after the potential shift, indicating a rapid phase transition of Pd and Pd₄Ag at this potential. A consequent plateau, presumably correlated to an equilibrium state, was reached after a period of electrolysis. The phase transition response at 100 s in each electrolyte is converted into the in situ H/M ratio (see details in Fig. S14), as shown in Fig. 4c. In the absence of CO₂RR, the formation of subsurface hydride ($MH_{subsurface}$) is in che-mical equilibrium with surface adsorbed hydrogen ($H_{ads}$), as shown in Eq. 10. This equilibrium could be significantly shifted by the H⁺ con-centration (i.e., pH) in the electrolyte. As a result, the H/M ratio in Pd was considerably higher when in strong acid HClO₄, as compared to KHCO₃ and K₂HPO₄/KH₂PO₄ environments (see red bars in Fig. 4c). Interestingly, a similar trend was not observed in Pd₄Ag, demonstrat-ing the unique H sorption thermodynamics in response to the H⁺

concentration in this alloying structure (blue bars in Fig. 4c). First, due to the alternation in the electronic structure of Pd and the proximity effect after Ag doping, the H binding energy is reduced on $Pd_4Ag$ surface, leading to the lower H/M ratio in $Pd_4Ag$ as compared to pure Pd (Fig. 4c). Similar conclusions have also been reached in dynamic ETS-CV investigations (Fig. 2). Second, when acidic $HClO_4$ was switched to more neutral electrolytes such as $KHCO_3$ and $K_2HPO_4/KH_2PO_4$, no significant decrease in the H/M ratio in $Pd_4Ag$ was observed (in sharp contrast to Pd). This is probably due to the influence of pH on the hydrogen binding energy (HBE) of surface $H_{ads}$. Zhu et al. demonstrated that the HBE on Pt is increased at low pH through in situ surface-enhanced infrared absorption spectroscopy (SEIRAS), which contributes to its higher HER activity[58]. This conclusion theoretically applies to Pd as the electrolyte-dependent HER[1] or other hydrogenation[59] activities can also be observed on Pd and other metals. As a result, while a low $H^+$ concentration (from $HClO_4$ to $KHCO_3$ and $K_2HPO_4/KH_2PO_4$) tends to shift the equilibrium in Eq. 10 to the left, reducing the hydrogen concentration in both $M-H_{ads}$ and $M-H_{subsurface}$ phases, the concurrently reduced HBE only on surface $H_{ads}$ facilitates the hydrogen diffusion into the subsurface Pd lattice[54], promoting $M-H_{subsurface}$ formation. Therefore, the influence of electrolyte can cast different influences on the sub-surface hydride formation in opposite directions. For $Pd_4Ag$ ($Pd^{\delta-}$) system that intrinsically has smaller HBE than Pd, the electrolyte-induced alternation in surface HBE plays a more critical role that is significant enough to counterpart the influence from $H^+$ concentration, thus leading to a more balanced H soption kinetics in different electrolytes as revealed in Fig. 4c. The similar phenomenon was also demonstrated in a PdPt alloy by in situ XAS and XRD, where substantially strong HBE on surface Pt inhibited the subsurface H diffusion[28]. For further confirmation, we have also conducted same ETS investigations on pure Au, showing that for metals with considerably weak HBE, the sub-surface hydride formation can only be observed in $KHCO_3$ rather than $HClO_4$ (Fig. S24). Overall, our ETS-derived in situ H/M ratios effectively reveal the role of electrolyte in surface HBE for Pd-based catalysts.

Under CO2RR conditions, much higher H/M ratios in both $KHCO_3$ and $K_2HPO_4/KH_2PO_4$ were observed in the pure Pd system (Fig. 4c), which are not completely consistent with the dynamic CV-ETS results shown in Fig. 3. The equilibrium H/M ratio in these near neutral electrolytes follows a sequence: $K_2HPO_4/KH_2PO_4$ ($CO_2$) > $KHCO_3$ ($CO_2$) > $K_2HPO_4/KH_2PO_4$ (Ar) > $KHCO_3$ (Ar) (indicated by the arrow in Fig. 4c), which is highly correlated with the pH sequence of electrolytes (8.3 > 7.2 > 6.8 > 6.3). Different to the extreme low pH in the acidic electrolyte ($HClO_4$), we speculate that the pH effect on HBE of surface $H_{ads}$ is similar in these four buffered electrolytes, and the difference in proton-donating capacity indicated by the different $pK_a$ values of proton donors (Fig. 4c) is critical to the formation of $H_{ads}$ and $MH_{subsurface}$. In this case, $H^+$ in Eq. 10 can be replaced by proton-donating $HA^{x-}$, and the $pK_a$ value of a buffer electrolyte can serve as a better descriptor for the evaluation of electrolyte-modulated subsurface phase transition, as shown in Fig. 4c. It can be noted that the same trend in the H/M ratio-electrolyte $pK_a$ correlation is observed in $Pd_4Ag$ ($Pd^{\delta-}$) system, indicating the critical role of proton-donating capacities of local electrolyte environments. Meanwhile, the overall subsurface H/M level in $Pd_4Ag$ is still below Pd under CO2RR, revealing the actual equilibrium state under operando electrolysis, again providing the direct evidence that the general HBE (both for $H_{ads}$ and $M-H_{subsurface}$) is relatively reduced in $Pd_4Ag$, even with comprehensive surface/subsurface physical chemical processes. In addition, the H/M ratio in both Pd and $Pd_4Ag$ under CO2RR conditions are even higher than in $HClO_4$, which highlights the presence of local (within electrochemical double layer) equilibrium between $CO_2$ and $H_2CO_3$ (Eq. 2) during CO2RR.

The transient ETS measurement also allowed for the investigation of time-dependent phase transition kinetics. The $t_{50}$ and $t_{90}$ (time for

50% and 90% level of maximum phase transition) in different electrolytes are subtracted and summarized in Fig. 4d. Both Pd and $Pd_4Ag$ demonstrated a fast kinetics to reach equilibrium hydride formation in $HClO_4$ (<0.1 s), which originates from the same reason that accounts for high equilibrium H/M ratios: the high $H^+$ concentration and high HBE for surface $H_{ads}$ under strong acidic electrolytes (also see Fig. S25). In $KHCO_3$ and $K_2HPO_4/KH_2PO_4$, much lower $H^+$ concentration certainly slows down the H sorption kinetics ($t_{50}$ between 0.4–5 s) despite the compensation from proton-donating anions that maintains the equilibrium states, clearly differentiating the H sorption kinetics with two different hydrogen sources ($H_3O^+$ and $HA^{x-}$). The change of H sorption kinetics under CO2RR shows consistent conclusions with dynamic CV-ETS tests, the addition of $CO_2$ slows down the hydride formation in $KHCO_3$ (Fig. 4d and Fig. 3e, f) while accelerates it in $K_2HPO_4/KH_2PO_4$ (Fig. 4d and Fig. 3i, j). These transient results again demonstrate the comprehensive H sorption kinetics under the balance between CO2RR inhibition (competing for surface $H_{ads}$) and proton donation from surrounding electrolytes (including $H_2CO_3$ formed in near-surface $CO_2$-electrolyte equilibrium). Furthermore, it is evident that the transient H sorption kinetics of $Pd_4Ag$ is overall slower than that of pure Pd, which is more obvious in $t_{90}$, further confirming the weak $M-H$ interaction in $Pd_4Ag$ from the kinetic point of view.

## Connection between CO2RR performances and H sorption processes

The dynamic, equilibrium and transient quantification of the subsurface H/M level (and therefore the H sorption kinetics) can be correlated to the CO2RR performance under each specific condition. We first performed the density functional theory (DFT) calculations (see Fig. S13) to rationalize the reduced CO poisoning and high formate production activity on $Pd_4Ag$ surface (as compared to pure Pd), demonstrating the role of charge transfer and geometric effects on the reduced $M-H$ interaction and CO poisoning[20]. In addition, we found that both Pd and $Pd_4Ag$ exhibit different CO2RR performances in varying H-donating electrolytes (Fig. 3a–c). $K_2HPO_4/KH_2PO_4$ can accelerate the H sorption kinetics, which is reflected by the increased phase transition degree (H/M ratio) in $Pd_4Ag$ (from 0.38 in $KHCO_3$ to 0.48 in $KH_2PO_4/K_2HPO_4$) during CO2RR (see Fig. 4c and Table S2). Correspondingly, an increase in formate FE of about 8% on $Pd_4Ag$ was obtained at $-0.4$–$-0.2$ $V_{RHE}$ (Fig. 3a), proving that strong proton supply in $KH_2PO_4/K_2HPO_4$ electrolyte and high degree in operando phase transition (H doping) would be beneficial for reducing CO poisoning and promoting formate production[18,24,28] (Fig. 3b). As a result, $Pd_4Ag$ demonstrated a significantly enhanced current density and stability for formate production compared with pure Pd (Fig. 3a, b). It should also be mentioned that the enhanced proton reduction kinetics in $KH_2PO_4/K_2HPO_4$, as revealed by ETS investigation, simultaneously promotes the competing HER (Fig. 3c)[50,51] and thus leads to obviously diminished formate FEs (Fig. 3a) at low overpotential region ($-0.15$–$-0$ $V_{RHE}$), where CO poisoning and CO2RR is relatively weak and less dominant. As a comparison, the H/M ratios of pure Pd in the $CO_2$-saturated $KHCO_3$ and $KH_2PO_4/K_2HPO_4$ (0.48 and 0.51) are close due to its originally strong $Pd-H$ interaction, and $KH_2PO_4/K_2HPO_4$ mostly accelerates the HER activity and the formate FEs are thus reduced (Fig. 3a). Therefore, both CO2RR and HER performances can be modulated by electrolyte environments that fundamentally determines the H sorption equilibrium and kinetics, and it is important to strike a balance among different and comprehensively connected near-surface, on-surface and sub-surface reactions, as summarized in Fig. 4e.

In summary, in situ electrical transport measurements were utilized to monitor and quantify the H sorption processes during CO2RR in either metallic (Pd, Pt, and Au) or bimetallic ($Pd_4Ag$) catalytic materials under scanning voltametric (dynamic) or potentiostatic (steady-state) electrolysis conditions. While fast subsurface hydride formations were observed in strong acidic environments, the proton-

donating capacity (indicated by the p$K_a$) of near neutral electrolytes, including the near-surface $H_2CO_3$ in equilibrium with $CO_2$, was found to serve as a key impacting factor to the H sorption kinetics and $CO_2$RR performances. Compared to pure Pd, more negative phase transition potentials and lower equilibrium H/M ratios were observed in bimetallic $Pd_4Ag$, providing direct evidence for the weakened M−H interaction and origin of promoted $CO_2$RR in alloy catalysts. The high degree of in situ M−H$_{subsurface}$ along with the fast H sorption kinetics would also be beneficial for reducing CO poisoning and promoting formate production. The H sorption features identified and quantified in various electrolyte environments under operando reaction conditions therefore provide a general and effective platform for the mechanistic investigation of electrocatalytic reactions involving hydrogenation processes.

## Methods

### Chemicals
Silver nitrate ($AgNO_3$, 99.98%), Ethylene glycol (EG, 99%), and palladium chloride ($PdCl_2$, 60% Pd basis) were purchased from Sigma-Aldrich. Ascorbic acid (99.98%), potassium hydrogen carbonate ($KHCO_3$, 99.7%), perchloric acid ($HClO_4$, 48–50%), n-butyl alcohol (99.4%), and nafion solution (5 wt%) were purchased from Alfa Aesar. Dihexadecyldimethylammonium chloride (75%) was purchased from Beijing Warwick Chemical Co., Ltd. Potassium hydroxide (KOH, 99.999% metal trace) was purchased from Innochem. Sodium bicarbonate ($NaHCO_3$, 99.8%), potassium phosphate dibasic anhydrous ($K_2HPO_4$, 99.0%), potassium phosphate monobasic ($KH_2PO_4$, 99.95%), and sodium perchlorate ($NaClO_4$, 99%) were purchased from Aladdin. Chloroauric acid ($HAuCl_4$, 99%) and chloroplatinic acid ($H_2PtCl_6$, 99.9%) were purchased from AdamasBeta. Ethanol (EtOH, 99.7%), isopropanol (99.7%), and N,N-Dimethylformamide (DMF, 99.5%) were purchased from Sinopharm Chemical Reagent Co. Ltd. (Shanghai, China). Sodium citrate dihydrate ($Na_3Citrate·2H_2O$, 99.0%) was purchased from Macklin. Polyvinyl pyrrolidone (PVP, MW=30000) was purchased from Shanghai yuanye Bio-Technology Co., Ltd. Sodium iodide (NaI, 99.2%) was purchased from Bide Pharmatech Ltd. HCl (36-38%) was purchased from Yonghua Chemical Co., Ltd. To prepare 10 mM $H_2PdCl_4$ solution, 0.355 g of $PdCl_2$ was dissolved with 20 mL of 0.2 M HCl solution in a 200 mL volumetric flask and further diluted to 200 mL by deionized (DI) $H_2O$. $K_2HPO_4$/$KH_2PO_4$ electrolyte was prepared by dissolving $K_2HPO_4$ and $KH_2PO_4$ in DI $H_2O$ with total concentration of 0.1 M and a molar ratio of 72/28. All chemicals were used without further purification.

### Synthesis of Pd nanowires
Pd nanowires were synthesized using a previously reported method[38]. Typically, $PdCl_2$ (17.7 mg), NaI (300 mg), PVP (800 mg), and 12.0 mL DI $H_2O$ were mixed, followed by ultrasonic treatment for 30 min and magnetic stirring for 12 h. The resulting homogeneous solution was then transferred to a 25 mL Teflon-lined autoclave and then maintained at 200 °C for 2 h. After cooling down to -25°C, the products were cleaned with ethanol and DI $H_2O$ for several times and dispersed in 4 mL DI $H_2O$ for further use.

### Synthesis of $Pd_4Ag$ nanowires
$Pd_4Ag$ nanowires were synthesized using a previously reported method[20]. Typically, 0.1 mM of dihexadecyldimethylammonium chloride was dissolved in 10 mL of DI $H_2O$ in a round-bottom flask, heated to and kept at 95 °C for 30 min. It was added with 2 mL of 10 mM $AgNO_3$ and 8 mL of 10 mM $H_2PdCl_4$ to form a homogeneous solution, and then with 1 mL of freshly prepared 0.3 M ascorbic acid solution to initiate the co-reduction of $Pd^{2+}$ and $Ag^+$. After 40 min continuous reaction, the solution was naturally cooled down to -25°C. The final product was cleaned with isopropanol, ethanol and DI $H_2O$ for several times and dispersed in 2 mL ethanol for further use.

### Synthesis of Pt nanowires
Pt nanowires were synthesized using a previously reported method with slight modification[37]. Typically, 0.1 M $H_2PtCl_6$ aqueous solution (400 μL) was added to a mixed solvent containing 12 mL EG, 12 mL DMF, and 700 mg KOH. After stirring for 30 min, the resulting solution was transferred into a 50 mL Teflon-lined autoclave and then maintained at 170 °C for 8 h. After cooling down to -25 °C, the products were cleaned with ethanol and DI $H_2O$ for several times and then dispersed in 4 mL ethanol for further use.

### Synthesis of Au nanoparticles
Au nanoparticles were synthesized using a previously reported method with slight modification[60]. Typically, 51.5 mg $Na_3Citrate·2H_2O$ was dissolved in 200 mL DI water, and the mixture was then heated to boiling in an oil bath under magnetic stirring. 0.1 M $HAuCl_4$ aqueous solution (500 μL) was then added into the $Na_3Citrate$ solution and the resulting mixture was kept in boiling for 10 min. After cooling down to -25 °C, the products were collected by centrifugation and finally dispersed in 4 mL EtOH for further use.

### Characterizations
XRD were conducted on a PANalytical X-ray diffractometer with a Cu Kα resource. XPS spectra were carried out on an Ultra DLD XPS spectrometer. The morphology of the as-synthesized nanomaterials were measured by TEM (JEM-2100). The morphology and thicknesses of on-chip films were measured by SEM (Hitachi S-4800) and AFM (Bruker Dimension Icon). The pH of electrolytes were measured by a pH meter (PB-10, Sartorius). ICP analysis was carried out on Aurora M90 inductively coupled plasma optical emission spectrometer.

### Electrochemical $CO_2$RR measurements
$CO_2$RR measurements were performed at the temperature range of 23–25 °C in a custom-designed H-cell using the standard three-electrode system. The working electrode was a catalyst-loaded glassy carbon plate with a size of $1 \times 1$ cm². For its preparation, 1 mg of catalyst powder, 0.5 mg of Ketjenblack carbon, and 6 μL of Nafion solution (5 wt%) were dispersed in 250 μL of ethanol, and subjected to ultrasonication for 30 min to form a homogeneous ink. The ink was then drop-casted onto the glassy carbon plate to achieve a catalyst areal loading of 1 mg/cm². The working electrode and a saturated calomel reference electrode (SCE) were located at the cathodic compartment; a graphitic rod as the counter electrode was located at the anodic compartment. These two compartments were each filled with 25 mL of 0.1 M $KHCO_3$ or 0.1 M $K_2HPO_4$/$KH_2PO_4$ (n($K_2HPO_4$)/n($KH_2PO_4$) = 72/28), and separated by a Nafion membrane. Before $CO_2$RR measurements, the electrolyte was bubbled with high-purity $CO_2$ (>99.999%) at 20 sccm for >40 min. During $CO_2$RR measurements, the gas flow was maintained to ensure the $CO_2$ saturation of the electrolyte. All potential readings were recorded in SCE and subsequently converted to RHE with necessary iR compensation. Chronoamperometric (i−t) curves were collected at selected potentials between −0.5–0 $V_{RHE}$. Formate in the catholyte from $CO_2$RR was analyzed using an ion chromatograph (Dionex ICS-600).

### Preparation of free-standing films from nanowires and nanoparticles
The films were prepared using a previously reported method with slight modification[35,40]. In a typical self-assembly experiment for nanowires or nanoparticles films, 1 mL as-prepared EtOH dispersion of $Pd_4Ag$, Pt, or Au was mixed with 2 mL $H_2O$, followed by ultrasonication for 5 min. 2.5 mL n-butanol was then added into the mixture, followed by ultrasonication for another 1 min. The final suspension was then added drop by drop into a flask (about 7.5 cm in diameter) filled with Milli-Q water. A piece of catalyst film was finally formed at the water/air interface and was later transferred onto the Si wafer.

## Fabrication of the devices

Devices were fabricated using a previously reported method[35], as schematically illustrated in Fig. S4. Typically, Si wafer (p++ with 300 nm thermal oxide) with pre-patterned Au electrodes (Ti/Au, 20 nm/50 nm) was used as substrate. A poly(methyl methacrylate) (PMMA, A8, MicroChem Corp.) film (~1000 nm thick) was spin-coated on the Si wafer, and consequent E-beam lithography (EBL) was used to open windows with defined shape. The pre-prepared (by co-solvent evaporation) free-standing film of Pd$_4$Ag, Pt, or Au (on the water surface) was then transferred into the window. To obtain the on-chip films with different thicknesses, the transfer process is repeated for 1–3 times. For a fine-controlled deposition, the substrate was treated by oxygen plasma before deposition. Due to the presence of a small amount of surfactant on the surface of nanowires even after centrifugation and cleaning, the preparation of free-standing films from the as-prepared Pd nanowires is hard to proceed. The Pd devices were fabricated by direct drop-casting of the Pd nanowires (dispersed in water) into the PMMA window. To obtain Pd devices with thin thicknesses, the dispersion of Pd nanowires were diluted by 6 or 24 times. After the removal of PMMA template, nanowires were then patterned on the device substrate with desired designs. To eliminate the influence of electrolyte and to avoid electrochemical reactions on the metal electrodes, another layer of PMMA (~500 nm thick, electrochemically inert) was then deposited on device with spin-coating. A smaller window that only exposes catalysts was opened by e-beam lithography. A drop (2 μL) of Nafion solution (0.1 wt% in EtOH) was added on the top of the sample to ensure its mechanical stability during electrochemical reactions. The final device, with exposed catalysts and PMMA-protected electrodes was used for on-chip electrochemistry and in situ electrical transport spectroscopy.

## Concurrent voltammetry (CV) and conductance (ETS) measurements

Concurrent CV and ETS measurements were performed at the range of 23–25 °C, using a two-channel source-measure unit (SMU, Keysight B2902a). A first SMU channel was used as a potentiostat to perform the on-chip CV by applying the potential ($V_G$) of source/drain electrode (acting as working electrode) as to the reference electrode (leak-free Ag/AgCl), while collecting the current ($I_G$) through the counter electrode (platinum wire). A second SMU channel was used to record ETS signals by supplying a small bias potential (50 mV) between source and drain electrodes and collecting the electrically conductive current ($I_{SD}$). For a typical measurement in this study, the Gate (Faradaic) current approaches several microamperes. Therefore, the in-device CV current may affect the ETS current and a background subtraction is needed before the data analysis (Fig. S3).

## Computational method

The DFT calculations were performed using the revised Perdew-Burke-Ernzerhof functionals (RPBE)[61] of generalized gradient approximation (GGA) implemented in the Vienna Ab-initio Simulation Package (VASP) code[62,63]. The projector-augmented wave (PAW) method[64,65] was applied to describe the electron-ion interactions. A kinetic energy cutoff for the plane wave expansions was set to be 520 eV. The method of Methfessel-Paxton (MP) was applied and the width of the smearing was chosen as 0.2 eV. The supercell of ($\sqrt{13} \times \sqrt{7}$) R19° with five atomic layers was chosen to construct the Pd (111) surface, and two Pd atoms were replaced by Ag atoms in each atomic layer to construct the Pd$_4$Ag surface. More than 10 Å of vacuum space was used to avoid the interaction of the adjacent images. For sampling the reciprocal space, **k**-points of Γ-centered $4 \times 3 \times 1$ were used for surface calculations. All structures were fully relaxed until the force components were less than 0.03 eV·Å$^{-1}$. Implicit solvent model was used in our calculations by VASPsol[66,67]. A Debye screening length of 9.61 Å was chosen, which corresponds to a bulk ion concentration of 0.1 M. The non-

electrostatic parameter, TAU, was set to zero for the purpose of convergence.

The adsorption energy of CO is defined as:

$$E_{b,CO} = E^*_{CO} - E^* - E_{CO}$$

where $E_{*CO}$, $E_*$ and $E_{CO}$ are the total energy of the surface with adsorbed CO, pristine surface, and CO molecule in the gas phase, respectively.

The adsorption energy of H is defined as:

$$E_{b,H} = E^*_{H} - E^* - 1/2E_{H2}$$

where $E_{*H}$, $E_*$, and $E_{H2}$ are the total energy of the surface with adsorbed H, pristine surface and H$_2$ molecule in the gas phase, respectively.

## Data availability

The data supporting the conclusions of this study are available within the paper and its Supplementary Information. Additional data are available from the corresponding author upon reasonable request. Source data for Figs. 2–4 are provided with this paper. Source data are provided with this paper.

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

## Acknowledgements

We thank Xinnan Mao and Dr. Lu Wang for the assistance in DFT calculations. We acknowledge the support by the Natural Science Foundation of China (Project No. 22172075 and No. 92156024 to M.D.), the Fundamental Research Funds for the Central Universities in China (Project No. 14380273 and No. 0210/14380174 to M.D.), Natural Science Foundation of Jiangsu Province (Project No. BK20220069 to M.D.), Beijing National Laboratory for Molecular Sciences (Project No. BNLMS202107 to M.D.) and National Natural Science Foundation of China (Project No. 21902114 to N.H.).

## Author contributions

M.D. and Z.M. designed the research. M.D., Y.L., Z.M., and N.H. supervised the research. Z.M. carried out the device fabrication, ETS measurements, and investigations. N.H. carried out material preparations and $CO_2RR$ measurements. Z.M. and M.D. analyzed the data. D.X., B.T., F.W., Y.W., Y.S., C.L., and X.W. contributed to characterization of catalysts. P.Z. provided instrumental assistance for the micro-fabrication. Z.M., N.H., Y.L., and M.D. co-wrote the paper. All authors have read and commented on the manuscript.

## Competing interests

The authors declare no competing interests.
