## [Peer Review File · Nature Communications]

REVIEWER COMMENTS

Reviewer #1 (Remarks to the Author):

The authors of this study combine voltammetry with electrical transport measurements to add to our understanding of the role of adsorbed and absorbed hydrogen and solution phase proton-donors in carbon dioxide electroreduction.

I believe this work is well performed, timely, and provides a very interesting set of results to the electrocatalysis community. However, I would encourage the authors to address the following concerns to improve their paper.

1) To be honest, the bulk of the results and discussion about Figures 3, 4, and 5 are a bit difficult to read. I don't think it requires significant changes, but minor tweaks for clarity are necessary. It seems the authors have a lot of points they want to make, and it is difficult to follow which are most important.

If the authors believe understanding the role of proton-donating species is important - I would consider moving Figure S21 to the main text - this may make it easier to understand what one can conclude from Figures 3 and 4.

Also - I would recommend the authors consider adding some kind of summarizing figure which highlights their key conclusions.

The authors may want to streamline their explanations of why they think certain things are occurring. (My suggestion would be to focus on what you can most directly conclude from your measurements, then put in context with prior work. I get the impression that with every measurement comes a discussion of how this fits perfectly into a story with prior work - may be better to discuss more of your results collectively first, then build/add to your story into what is known already). For what its worth --- the Conclusion section is clear + easy to read.

2) In Figure 3 - the authors attribute the large hysteresis in the electrical transport measurements (and therefore in hydrogen absorption) to be due to the slow kinetics of hydrogen reduction from water in a neutral/alkaline electrolyte. Did the authors perform any additional measurements to corroborate this? Namely, were CV's and ETS measurements performed at varying scan rates? If the hysteresis is scan rate dependent, this would pretty strongly support it is a kinetic issue. (Are the results in Figure S21 enough to corroborate this?).

3) Does the absorption of hydrogen into the Pd4Ag alloy cause a segregation of the constituents (de-alloying)?

4) Paper could use minor copy-editing to improve English writing.

Reviewer #2 (Remarks to the Author):

The manuscript titled "Critical role of hydrogen sorption kinetics in electrocatalytic CO₂ reduction revealed by on-chip in situ transport investigations" Mu et al. probes the hydrogen sorption properties of Pd and Pd₄Ag alloys using a novel electron transport spectroscopy method. The authors detail the role of different electrolytes on the hydrogen sorption properties of these materials as well as how CO₂ influences hydrogen sorption and connect their observations to trends in electrolyte pK_a, pH, and CO₂ content. Overall, this investigation is interesting and their methods should be broadly applicable to many different systems of high complexity, however, possible diffusion-related artifacts as well as a lack of computational experiment to substantiate their claims prevent me from recommending this publication in Nature Communications. In addition, the manuscript has several language issues, which, in addition to the somewhat unfocused nature of the text, make the article difficult to read.

Major Points:

(1) Diffusion of the electrolyte to the surface of the electrode is potentially problematic for the data presented. For example, significant weight of the authors' claims is given to the absolute electrochemical potentials of the Habs and Hads, however, the authors clearly show in figure S9 that due to diffusion issues, these potentials shift by hundreds of mV depending on the electrode thickness– not a material-dependent property. In addition, the authors present time-resolved ETS in different electrolytes with and without CO₂ present. Both the 'equilibrium potentials' and the reaction kinetics (as defined by the half-life of a transient current signal) change based on the electrolyte and greatly influence the authors' claims. Given the sensitivity of measurement on the sample thickness and presumably sample morphology, the authors need to show that these ETS and CV measurements reflect the Pd and Pd₄Ag, and not the morphology or structure of the sample. This can be accomplished by supplementing their current data with additional ETS and CV curves of different sample thicknesses in the H₂CO₃ and H₃PO₄ electrolytes similar to data presented in Figure S9. In addition, scan-rate dependence data on the peak positions of hydrogen absorption/desorption would be valuable.

(2) The authors make several claims that the electronic structure of the Pd₄Ag alloy is responsible for both the reduced CO poisoning and the reduced H absorption capacity. Based on their data, however, a simple geometric effect can explain their observations. The stoichiometry of Pd₄Ag₁ implies that, in a homogeneous alloy, every palladium atom in an fcc lattice is in contact with a Ag atom. Ag is known to both rapidly evolve CO (due to the low CO binding energy) and does not absorb hydrogen into its lattice. A silver atom near every Pd may therefore inhibit CO poisoning and H absorption–based only on its proximity and not on orbital hybridization. If the authors are going to claim electronic structure is the origin of their observation, they need to supplement their data with additional computational evidence showing such an effect.

Minor Points:

- (1) Figure 2, panel a,b,c. Please clearly indicate which y-axis corresponds to the plotted data. As it currently stands, it is very difficult for the reader to interpret these plots.
- (2) Figure S15 panels a and c. The arrows seem to be going in opposite directions with increasing scan. Please alter the figure to clearly indicate the scan direction.
- (3) In the text and figure 4a, the authors switch notation between the empirical chemical formula and an abbreviated name. Specifically, H₂CO₃ and 'PBS'. To improve the readability of the figures and text, the authors should choose either the chemical formula or abbreviations, but not both.

Reviewer #1:

The authors of this study combine voltammetry with electrical transport measurements to add to our understanding of the role of adsorbed and absorbed hydrogen and solution phase proton-donors in carbon dioxide electroreduction. I believe this work is well performed, timely, and provides a very interesting set of results to the electrocatalysis community. However, I would encourage the authors to address the following concerns to improve their paper.

Reply: We greatly appreciate the reviewer's positive comments on our study, and thank for the insightful suggestions that have inspired us to conduct key measurements on the subject and to improve the clarity in writing this manuscript. Per reviewer's suggestions, we have added additional data along with more concise and clarified discussions, which we believe have further improved the quality of this paper.

(1) To be honest, the bulk of the results and discussion about Figures 3, 4, and 5 are a bit difficult to read. I don't think it requires significant changes, but minor tweaks for clarity are necessary. It seems the authors have a lot of points they want to make, and it is difficult to follow which are most important. If the authors believe understanding the role of proton-donating species is important - I would consider moving Figure S21 to the main text - this may make it easier to understand what one can conclude from Figures 3 and 4. Also - I would recommend the authors consider adding some kind of summarizing figure which highlights their key conclusions. The authors may want to streamline their explanations of why they think certain things are occurring. (My suggestion would be to focus on what you can most directly conclude from your measurements, then put in context with prior work. I get the impression that with every measurement comes a discussion of how this fits perfectly into a story with prior work - may be better to discuss more of your results collectively first, then build/add to your story into what is known already). For what its worth --- the Conclusion section is clear + easy to read.

Reply: We thank the reviewer for pointing out the critical issue of unclear presentation of results in the manuscript. In this study, we have indeed devoted a great deal of space in the text to explain what things are occurring, partially due to the reason that this is the first time ETS is conducted for investigating H sorption processes and CO₂RR. We now realize that it is a bit too much and makes the main text difficult to read. Per reviewer's first suggestion, we have merged original Figure 3, Figure 4 and Figure 5f-h as the **new Figure 3**, together with a newly constructed logic of discussions corresponding to this major piece of data in the revised manuscript, as the following:

First, in **new Figure 3a-c**, we show the CO₂RR performances from Pd/Pd₄Ag under different reaction conditions, demonstrating the impact of different proton-donating electrolytes on the CO₂RR performances, and how Pd and Pd₄Ag respond differently to these conditions. These data lead to the introduction of the critical role of H sorption kinetics in Pd-catalyzed CO₂RR processes.

Second, in **new Figure 3d**, we further establish the connections between the H sorption kinetics and the corresponding ETS signals in the extended electrolytes (starting from KHCO₃), which also serves as an effort to address the **question #2** from the reviewer. On this basis, in **new Figure 3e-h**, we demonstrate the ETS studies under CO₂RR conditions to elucidate the H sorption kinetics during CO₂RR and rationalize the distinct CO₂RR performance of different Pd-based catalysts in KHCO₃. Specifically in this big section, we also rearranged the original discussion paragraphs in the revised manuscript: we first conclude the competition between CO₂RR and

hydrogen sorptions in KHCO_3 directly from the ETS results, and then discuss the possible reactions on the surface that is “in the context with prior work”. The ETS results in a different electrolyte ($\text{K}_2\text{HPO}_4/\text{KH}_2\text{PO}_4$) were then moved forward in the revised manuscript as new Figure 3i-l, as an effort to first “discuss more of our results collectively”. Consequently, new Figure 3m and new Figure 3n were added in the revised manuscript as the suggested “summarizing figure which highlights the key conclusions”, for the intuitive understanding of interfacial H sorption processes during CO_2RR , different M–H interactions within the catalysts, and proton-donating capacities of electrolytes as concluded from electrochemical and ETS investigations. Overall, thanks for the valuable and detailed instructions by the reviewer, we believe the revised manuscript now have more clarified logic, with the addition/rearrangement in the new Figure 3 and the correspondingly updated discussions, we sincerely thank the reviewer for these suggestions.

Next, we moved original Figure S21 into the main text as new Figure 4d according to the reviewer’s other suggestion, to better clarify the H sorption kinetics. Due to the importance of this data, we have further conducted repeating tests and added error bars in new Figure 4c-d, to demonstrate the reproducibility and reliability of this data.

Overall, new Figure 3 and new Figure 4 were updated in the revised manuscript, with the edited/rearranged discussions as the following:

Starting from page 9, line 23: “**Electrochemical CO_2RR performances and their kinetic dependence on H sorptions in buffered electrolytes.** Based on the identification and quantification of *in situ* H sorption processes in acidic electrolyte enabled by transport measurement, we next aim to study the H sorption impacts on Pd-catalyzed CO_2RR in a commonly used bicarbonate buffer electrolyte (KHCO_3), and a similar phosphate buffer electrolyte ($\text{K}_2\text{HPO}_4/\text{KH}_2\text{PO}_4$) for comparison. The CO_2RR product distributions were firstly obtained under constant potential electrolysis from -0.5 to $0.0 V_{\text{RHE}}$, as shown in Figure 3a-c.”

“Figure 3b further shows that Pd_4Ag has higher formate production rate than Pd at all tested potentials. The above results indicate an overall better CO_2RR performance after the alloying with Ag atoms.”

“Interestingly, when the electrolyte was switched to $\text{K}_2\text{HPO}_4/\text{KH}_2\text{PO}_4$, the overall formate FEs on pure Pd were reduced (Figure 3a). In comparison, Pd_4Ag showed even more complex changes in CO_2RR performance. The formate FEs also experienced an obvious decrease at low overpotentials (-0.15 to $-0 V_{\text{RHE}}$), yet a unique increase by $\sim 8\%$ was observed at high overpotential region (-0.4 to $-0.2 V_{\text{RHE}}$). Overall, the different CO_2RR performances in KHCO_3 and $\text{K}_2\text{HPO}_4/\text{KH}_2\text{PO}_4$ indicate distinct interfacial processes sensitive to the proton-donating electrolytes, which are presumably connected to the phase transition level of catalysts, CO poisoning/site blocking, formate production rates (Figure 3b) and HER kinetics (Figure 3c). It should be noted that the XRD after chronoamperometric studies reveal that H sorption processes during CO_2RR alloy does not cause a segregation (or any other structural or compositional change) in the Pd_4Ag (Figure S17), indicating its reversibility during the reaction.

ETS measurements were further conducted to elucidate the H sorption kinetics and rationalize the distinct CO_2RR performance of different Pd-based catalysts in different electrolyte environments. First, we have compared the H sorption kinetics by ETS in Ar-saturated KHCO_3 and HClO_4 at varying scan rates (Figure 3d&S18). In HClO_4 , significantly larger ETS hysteresis loops were observed to achieve complete H adsorption/desorption when the scan rate was increased from 10 to 80 mV/s, which confirms that the H sorption process is kinetic-dependent. In addition, while the degree of phase transition (corresponding to the ETS current level) kept unchanged in HClO_4 ,

lower degree of phase transition (dashed arrow in Figure 3d) was observed in KHCO_3 with the increasing scan rate. These results suggest the slow H sorption kinetics in near neutral electrolyte, which is reasonable due to the low concentration of hydronium ion and slow kinetics of water reduction in neutral/alkaline electrolytes⁴⁶⁻⁴⁹.

Figure 3e depicts the CV and ETS curves of Pd under CO_2RR conditions (in 0.1 M CO_2 -saturated KHCO_3). An obvious alternation in the H sorption hysteresis loop in ETS can be observed (marked with blue colored area and dashed arrow in Figure 3e) after the introduction of CO_2 , which may originate from the change of proton source and competitive surface reactions. Specifically, the addition of CO_2 increases the concentration of H_2CO_3 in the electrolyte...”

“The nonlinear variation of I_{SD} and phase transition with the negative shift of potential therefore strongly indicate the competition between CO_2RR and H sorption at potentials $< 0 V_{\text{RHE}}$ in KHCO_3 . Two parallel pathways were generally proposed for Pd-catalyzed CO_2RR , leading to formate or CO products²⁴:

in the formate or formic acid pathway:

in the CO pathway:

The formate mechanism involves a proton-coupled electron transfer (PCET) process, during which the proton transfer and electron transfer occur in a same elementary step, and the M–H bond is formed on the surface⁵²⁻⁵³. Although H adsorption on the surface is thermodynamically more favorable (by 0.33 eV) compared to subsurface H absorption (which leads to hydride formation), H could diffuse into the subsurface and then bulk fcc Pd lattice at more negative potentials^{18, 54-55}. The overall hydrogen sorption process within Pd and Pd_4Ag systems can be described by⁵⁶:

where the diffusion between H_{ads} and $\text{H}_{\text{subsurface}}$ follows the equilibrium that is determined by the chemical potentials of H atoms in each phase ($\mu_{\text{H}_{\text{ads}}}$ vs. $\mu_{\text{H}_{\text{subsurface}}}$). During active CO_2RR , the C1 intermediates occupy the Pd sites and inhibit the production of H_{ads} , shift the equilibrium between H_{ads} and $\text{H}_{\text{subsurface}}$, and eventually alter the level of hydride formation. In addition, the more favorable formate pathway (eq 4-6) will largely consume H_{ads} and slow down the kinetics of subsurface H diffusion and phase transition process. Importantly, these H-involving processes during CO_2RR can be reflected on the ETS signals corresponding to the H sorptions. On this basis, as the

unusual change in I_{SD} (Figure 3g) is in well correspondence with the high formate FE (>70%) in the potential range (>-0.2 V_{RHE}), our results therefore confirm that..."

On page 16, line 9: "As summarized in Figure 3m, while the H sorption kinetics is inhibited by CO₂RR-related H consumption and/or CO poisoning, it can be promoted by the local proton-donating species including H₂CO₃ in equilibrium with CO₂. It should also be noted that the H sorption is essentially a kinetic-dependent process, and the phase transition potentials may vary under different test conditions (potential scan rates, electrode geometries, etc.). Additional ETS tests indeed show positive shifts of the phase transition potential at smaller film thickness or slower scan rates (Figure S21), emphasizing the importance of consistence in test conditions. To this end, the ETS measurements conducted with 25 nm thin film thickness and 10 mV/s scan rate represent the experimental condition that gives close-to-intrinsic properties of the electrode materials, where the impact from the insufficient electrolyte diffusion to the sub-layer nanowires within the thin film device was minimized. With precise control of these experimental factors, the phase transition potential of Pd was determined to be about 150 mV more positive than that of Pd₄Ag in KHCO₃, reflecting the different M-H interactions. Additionally, by switching the electrolyte from KHCO₃ to K₂HPO₄/KH₂PO₄, the phase transition potentials of both Pd and Pd₄Ag markedly shift to 0~0.1 V_{RHE}, reflecting the strong proton-donation and fast H sorption kinetics in K₂HPO₄/KH₂PO₄. The key results and the corresponding conclusions were summarized in Figure 3n."

New Figure 3. CO₂RR and H sorption processes in KHCO₃ and K₂HPO₄/KH₂PO₄ electrolytes. a-c Potential-dependent selectivity for formate production (a), and current densities for formate production (b) and hydrogen evolution (c) with Pd (red) and Pd₄Ag (blue) catalysts in 0.1 M CO₂-saturated KHCO₃ (solid curves) and K₂HPO₄/KH₂PO₄ (dash curves). **d** Phase transition responses of Pd in 0.1 M Ar-saturated KHCO₃ and HClO₄ with varying scan rates. Film thickness is ~200 nm. **e-h** On-chip I_{SD}-V_G (ETS, solid) and I_G-V_G (CV, dash) curves of Pd (e, g) and Pd₄Ag (f, h) in 0.1 M Ar- and CO₂-saturated KHCO₃. **g** and **h** depict the enlarged negative-potential-sweeping ETS curves (0.2 to -0.5 V_{RHE}) in **e** and **f**, respectively. **i-l** On-chip I_{SD}-V_G (ETS, solid) and I_G-V_G (CV, dash) curves of Pd (i, k) and Pd₄Ag (j, l) in 0.1 M Ar- and CO₂-saturated K₂HPO₄/KH₂PO₄. **k** and **l** depict the enlarged negative-potential-sweeping ETS curves (0.2 to -0.5 V_{RHE}) in **i** and **j**, respectively. Solid arrows in **d-l** indicate the potential sweeping direction. Film thickness is ~200 nm. **m** Schematic illustration of different Pd-H states in KHCO₃ (left panel) and K₂HPO₄/KH₂PO₄ (right panel), and the corresponding CO₂RR processes at the interfaces. **n** Summary of phase transition potentials of Pd (blue) and Pd₄Ag (red) under CO₂RR conditions in KHCO₃ and K₂HPO₄/KH₂PO₄ obtained at 10 mV/s with different film thicknesses.

New Figure 4. Transient ETS quantifications of H sorption processes and CO₂RR performances during potentiostatic electrolysis. **a** Phase transition response of Pd and Pd₄Ag under potentiostatic conditions in different electrolytes (0.1 M). The potential was first kept at 0.5 V_{RHE} for 10 s and then shifted to -0.38 V_{RHE} for 100 s. Film thickness is ~25 nm. **b** Schematic illustration of M-H states and CO₂RR performances of Pd and Pd₄Ag in near neutral conditions. "PD" in **b** represents proton donor. **c** Summary of equilibrium H/M ratios obtained in different electrolytes at 100 s. The H/M ratios are obtained by first normalizing the responses in **a** with reference to the maximum phase transition responses (Pd: 45%, Pd₄Ag: 13%) obtained in 0.1 M HClO₄, and then referring to the published quantitative relationship between resistivity and H/M ratios of pure Pd and Pd₈₀Ag₂₀ alloy (Figure S13). On the top show the pH and pK_a values of the proton donors with relatively high proton-donating capacities. The pK_a of 0, 10.33, 7.21, 6.35 correspond to H₃O⁺, HCO₃⁻, H₂PO₄⁻ and H₂CO₃, respectively. **d** Summary of the time for 50% and 90% level of maximum phase transition in Pd and Pd₄Ag. **e** Proposed mechanism for proton donation, H sorption and CO₂RR on Pd-based materials in near neutral conditions. "M-int" in **e** represents the surface adsorbed intermediates during CO₂RR.

As other additional efforts to make the revised manuscript more clarified and easier to read, the "INTRODUCTION" part was slightly modified to better clarify the logic of this whole work, and several sub-titles were added in the "RESULTS AND DISCUSSION" section in the revised manuscript, as following:

In the Introduction: “Metal–hydrogen (M–H) interactions and the correlated chemical/catalytic hydrogen processes (H adsorption, absorption, evolution and oxidation) participated in multiple applications such as hydrogen fuel cell¹⁻², hydrogen/pH sensors³⁻⁶, metal hydride batteries⁷, and electrocatalytic hydrogen evolution reaction (HER) and hydrogenation reactions⁸⁻¹⁰. To this end, Palladium (Pd) is one of the mostly adapted materials, which serves as a typical model catalyst for the fundamental investigation of M–H states and dynamic hydrogen transitions, owing to its unique and rich interactions with hydrogen⁹⁻¹³. Among various Pd-catalyzed electrochemical reactions, **electrochemical carbon dioxide reduction (CO₂RR)** attracts most research attentions as it represents a sustainable means to reduce CO₂ emissions by converting it into valuable chemicals and hydrocarbon fuels, providing an effective and economical approach towards carbon neutralization¹⁴⁻¹⁷. **Numerous studies have been focused on the compositional and morphological innovations on Pd-based nanostructures¹⁸⁻²⁴ to obtain high current densities and Faradaic efficiencies (FEs) of desired products, and one particular effective approach is the use of bimetallic catalysts where alloying elements such as Ag²⁰⁻²¹ and Au²⁵⁻²⁶ can alter the electronic structure of Pd, regulate the intermediate adsorption energy, and finally improve the CO₂RR performance.** In analogy, H can be viewed as another alloying source for Pd: the formation of Pd–H bond involves a charge transfer process^{16, 18, 22, 24}, followed by the consequent transition to PdH_x as a separate phase. H atoms either adsorb on the surface or diffuse into the subsurface, and significantly alter the adsorption energy of reaction intermediates, such as *CO, HCOO* and *COOH^{18, 24}.

Despite the significant influence on the adsorption of intermediates in CO₂RR and other hydrogenation reactions, there are only few experimental approaches for the quantitative **measurement** of adsorbed/absorbed H atoms and corresponding H sorption kinetics in Pd-based catalysts under *operando* conditions. **Specifically**, the phase transition of Pd is buried at a solid/liquid interface, which is difficult for *in situ* characterizations and **poses** particular challenge in the study of corresponding electrocatalytic mechanisms. In most cases, *in situ* X-ray absorption spectroscopy (XAS) and *in situ* X-ray diffraction (XRD) were typically employed to characterize the Pd–Pd bond lengths and the lattice expansion during phase transition^{17-18, 22, 27-28}, **which successfully revealed the impact of catalyst morphologies on the potential range (with difference up to 100~300 mV) for PdH_x formation.** AC-impedance²⁹, quartz crystal microbalance³⁰ and cyclic voltammetry³¹⁻³³ are the commonly employed approaches for directly studying Pd–H interactions, however each individual methodology typically produces information on restricted dimension. To fully elucidate the comprehensive **electrocatalytic mechanisms that includes interfacial chemical processes and the local environments**, it is essential to bring up additional *in situ* approaches (better with alternative signaling mechanism) to complement existing characterization **toolbox** for the systematic investigation of Pd–H interactions and corresponding hydrogenation processes.”

In the main text, the following sub-titles were added: **“Catalyst preparation and device fabrication”**, **“ETS identification of *in situ* H sorption processes in perchloric acid”**, **“Electrochemical CO₂RR performances and their kinetic dependence on H sorptions in buffered electrolytes”**, **“Potentiostatic ETS analysis for near-equilibrium *operando* H sorption quantifications”**, **“Connection between CO₂RR performances and H sorption processes”**.

(2) In Figure 3, the authors attribute the large hysteresis in the electrical transport measurements (and therefore in hydrogen absorption) to be due to the slow kinetics of hydrogen reduction from water in a neutral/alkaline electrolyte. Did the authors perform any additional measurements to corroborate this? Namely, were CV's and ETS measurements performed at varying scan rates? If the hysteresis is scan rate dependent, this would pretty strongly support it is a kinetic issue. (Are the results in Figure S21 enough to corroborate this?).

Reply: We thank the reviewer for this inspiring suggestion on the ETS measurements at varying scan rates. This is certainly a classic approach used in CV tests to prove the kinetic-controlled electrochemical processes. Indeed, as hydrogen sorption includes the process of surface-adsorbed hydrogen atom gradually diffusing into the lattice, it will be correlated to the scan rate of potential.

The ETS of Pd were tested at varying scan rates in both Ar-saturated HClO₄ and KHCO₃, the results were added as new Figure 3d, new Figure S18 and new Figure S21 in revised MS and SI. In HClO₄, with the scan rates increased from 10 to 80 mV/s, larger hysteresis loops were indeed observed, which confirms that the H sorption process is kinetic-dependent. In addition, the I_{SD} in HClO₄ can eventually reach to a same level plateau (indicating same level phase transition degree) at all investigated scan rates, while in sharp contrast, lower degree of phase transition (indicated by the arrow in new Figure S18) is observed in KHCO₃ even at a more negative potential (-0.5 V_{RHE}) within the same range of scan rates. This difference further reveals the slow H sorption kinetics in near neutral electrolyte. In addition, more systematic ETS measurements on Pd/Pd₄Ag of varying film thicknesses, and with different potential scan rates, were added as new Figure S21, which further reveal the kinetic-dependent nature of the corresponding measurements (and help derive the key parameter intrinsic to the materials). Again, we thank the reviewer for the valuable suggestions. In summary, the ETS results obtained from different scan rates have been added as new Figure S18 and new Figure S21 in the revised SI, representative ETS results obtained at 10 mV/s and 80 mV/s have been added as new Figure 3d, and the corresponding descriptions have been updated in the revised manuscript:

Page 12, line 8: "ETS measurements were further conducted to elucidate the H sorption kinetics and rationalize the distinct CO₂RR performance of different Pd-based catalysts in different electrolyte environments. First, we have compared the H sorption kinetics by ETS in Ar-saturated KHCO₃ and HClO₄ at varying scan rates (Figure 3d & S18). In HClO₄, significantly larger ETS hysteresis loops were observed to achieve complete H adsorption/desorption when the scan rate was increased from 10 to 80 mV/s, which confirms that the H sorption process is kinetic-dependent. In addition, while the degree of phase transition (corresponding to the ETS current level) kept unchanged in HClO₄, lower degree of phase transition (dashed arrow in Figure 3d) was observed in KHCO₃ with the increasing scan rate. These results suggest the slow H sorption kinetics in near neutral electrolyte, which is reasonable due to the low concentration of hydronium ion and slow kinetics of water reduction in neutral/alkaline electrolytes⁴⁶⁻⁴⁹."

New Figure S18. I_{sp} - V_G (ETS) curves of Pd in 0.1 M Ar-saturated HClO_4 (a) and KHCO_3 (b) with different scan rates at 10, 20, 40 and 80 mV/s. Film thickness is ~ 200 nm. Dashed arrows indicate the influence of scan rate (i.e., dynamic reaction time) on the hysteresis and degree of Pd phase transition. Solid arrows in all figures indicate the potential sweeping directions.

New Figure S21. (a-d) Phase transition responses of Pd (a, b) and Pd_4Ag (c, d) in 0.1 M CO_2 -saturated KHCO_3 (a, c) and $\text{K}_2\text{HPO}_4/\text{KH}_2\text{PO}_4$ (b, d) with different scan rates and film thicknesses. (e-f) Summarized dependence of onset potential for the phase transition of Pd (e, f) and Pd_4Ag (g, h) on the potential scan rate with different film thicknesses.

(3) Does the absorption of hydrogen into the Pd₄Ag alloy cause a segregation of the constituents (de-alloying)?

Reply: We thank the reviewer for pointing out the possible issue of the structural change in the alloying catalyst during the hydrogen sorption processes or active electrocatalytic reactions. To address this issue, we have performed XRD measurements of Pd₄Ag before and after chronoamperometric electrocatalysis at -0.38 V (a typical potential for CO₂RR and H sorption) for 1 h (enough long compared with the time for ETS investigation) in 0.1 M CO₂-saturated KHCO₃ and K₂HPO₄/KH₂PO₄. As shown in **new Figure S17**, the diffraction peaks at 39.6°, 46.1° and 67.4° are consistent with those of the original Pd₄Ag powder sample. The peaks at 43.7° and 53.6° correspond to the (111) and (022) diffractions of graphite from the current collector. There is no obvious peak shift or generation of new diffraction peaks. These results support that the alloy structure of Pd₄Ag is well retained after electrolysis without phase segregation. The new data has been added as **new Figure S17** and the corresponding discussions have been updated in the revised manuscript, as following:

Page 10, line 25: “Overall, the different CO₂RR performances in KHCO₃ and K₂HPO₄/KH₂PO₄ indicate distinct interfacial processes sensitive to the proton-donating electrolytes, which are presumably connected to the phase transition level of catalysts, CO poisoning/site blocking, formate production rates (Figure 3b) and HER kinetics (Figure 3c). It should be noted that the XRD after chronoamperometric studies reveal that H sorption processes during CO₂RR alloy does not cause a segregation (or any other structural or compositional change) in the Pd₄Ag (Figure S17), indicating its reversibility during the reaction.”

New Figure S17. XRD patterns of the glassy carbon current collector (red curve), Pd₄Ag powders (black curve) and Pd₄Ag loaded on glassy carbon current collector after electrolysis at -0.38 V for 1 h in 0.1 M CO₂-saturated KHCO₃ (green curve) or K₂HPO₄/KH₂PO₄ (orange curve).

(4) Paper could use minor copy-editing to improve English writing.

Reply: We thank the reviewer for carefully reading the manuscript and pointing out the issue of English writing. We have gone through the manuscript and indeed found out many typographical, grammar mistakes and inappropriate expressions in the original manuscript, which have been fixed and highlighted in the revised manuscript. After the editing, we believe the quality of the English writing has been improved.

Reviewer #2:

The manuscript titled “Critical role of hydrogen sorption kinetics in electrocatalytic CO₂ reduction revealed by on-chip in situ transport investigations” Mu et al. probes the hydrogen sorption properties of Pd and Pd₄Ag alloys using a novel electron transport spectroscopy method. The authors detail the role of different electrolytes on the hydrogen sorption properties of these materials as well as how CO₂ influences hydrogen sorption and connect their observations to trends in electrolyte pK_a, pH, and CO₂ content. Overall, this investigation is interesting and their methods should be broadly applicable to many different systems of high complexity, however, possible diffusion-related artifacts as well as a lack of computational experiment to substantiate their claims prevent me from recommending this publication in Nature Communications. In addition, the manuscript has several language issues, which, in addition to the somewhat unfocused nature of the text, make the article difficult to read.

Reply: We greatly appreciate reviewer’s positive comments on the novelty/significance of our methodology and investigations, and thank for the valuable suggestions that have inspired us to include key measurements and additional computational results on the subject. In addition, we have made major changes in the revised manuscript to fix the typographical/grammar mistakes and to present a more concise and clarified discussion, as an effort to address the language/writing issue raised by the reviewer. Overall, we hope we can convince the reviewer that the additional experimental/computational data and modified writing have improved the quality of the revised manuscript, making it up to the high standard of this journal.

Major Points:

(1) Diffusion of the electrolyte to the surface of the electrode is potentially problematic for the data presented. For example, significant weight of the authors' claims is given to the absolute electrochemical potentials of the H_{abs} and H_{ads}, however, the authors clearly show in figure S9 that due to diffusion issues, these potentials shift by hundreds of mV depending on the electrode thickness– not a material-dependent property. In addition, the authors present time-resolved ETS in different electrolytes with and without CO₂ present. Both the ‘equilibrium potentials’ and the reaction kinetics (as defined by the half-life of a transient current signal) change based on the electrolyte and greatly influence the authors' claims. Given the sensitivity of measurement on the sample thickness and presumably sample morphology, the authors need to show that these ETS and CV measurements reflect the Pd and Pd₄Ag, and not the morphology or structure of the sample. This can be accomplished by supplementing their current data with additional ETS and CV curves of different sample thicknesses in the H₂CO₃ and H₃PO₄ electrolytes similar to data presented in Figure S9. In addition, scan-rate dependence data on the peak positions of hydrogen absorption desorption would be valuable.

Reply: We thank the reviewer for pointing out the critical issue of electrolyte diffusion in the ETS investigations, which is highly dependent on different test conditions including potential scan rate and thin film thickness rather than the intrinsic properties of electrode materials. This should certainly be systematically evaluated (through proper experimental design) to derive convincing measurement data and solid conclusions regarding the intrinsic properties of the electrode materials (Pd and Pd₄Ag).

To address this issue, we first tested the ETS of Pd in HClO₄ and KHCO₃ with varying scan rates, as suggested by the reviewer. The corresponding results were added as **new Figure 3d** and **new Figure S18** in revised MS and SI.

The large hysteresis in the ETS measurements (and therefore in hydrogen absorption) were linked to the slow kinetics of hydrogen reduction from water in a neutral/alkaline electrolyte in the original Figure 3. In the new scan-rate dependence measurements, with the scan rates increased from 10 to 80 mV/s, larger hysteresis loops were indeed observed in the ETS of Pd in HClO₄, which further confirm that the H sorption process is kinetic-dependent. In addition, the I_{SD} in HClO₄ can eventually reach to a same level plateau (indicating same level phase transition degree) at all investigated scan rates, while in sharp contrast, lower degree of phase transition (indicated by the arrow in new Figure S18) is observed in KHCO₃ even at a more negative potential ($-0.5 V_{RHE}$) within the same scan rate range. This difference further reveals the slow H sorption kinetics in near neutral electrolyte. Overall, it should be noted that the diffusion of the electrolyte has been recognized as an important process in the typical electrochemical voltametric measurement, and our original and new results certainly indicate it is also reflected on the ETS measurements. As a result, the scan-rate dependence data (such as hysteresis loops and the phase change degree) are indeed valuable to probe the H sorption dynamics in Pd-based system.

Second, on the basis of the significant impact of diffusion processes on the ETS measurement, the reviewer is also correct on the issue of variation in the absolute value of “onset potentials” for the phase change of Pd in the ETS tests, which seems to be influenced by the diffusion of electrolyte to the electrode surface (indicated by original Figure S9) and the diffusion of H atoms from the electrode surface into the subsurface lattice. To further address this issue, we further conducted systematic ETS measurements on both Pd and Pd₄Ag of varying film thicknesses, and with different potential scan rates, the new results were added as new Figure S21 in the revised SI. For pure Pd in KHCO₃, the phase transition potential is positively shifted by ~ 120 mV when film thickness was reduced from 200 nm to 25 nm (yellow rectangles in new Figure S21a). In addition, higher degree of phase transition and more positive phase transition potentials can be observed with the decreasing scan rate from 80 mV/s to 10 mV/s (yellow rectangle in new Figure S21a). Similar phenomenon can be observed in other tests conditions (yellow rectangles in new Figure S21a-d). All these results show that hydrogen sorption process is essentially a kinetic-dependent process, and slower scan rates and thinner film thicknesses would benefit the interfacial diffusion of electrolyte or hydrogen atoms. We would also like to emphasize here that, when the film thickness was reduced to ~ 25 nm, it corresponds to $\sim 1-2$ layers of nanowires, as confirmed by optical microscopy and AFM (Figure S8). In this case, the Pd NWs within the whole film were easily exposed to the electrolyte, and the influence from insufficient diffusion of electrolyte from top to underneath Pd NWs film was presumably reduced to a minimum level. Further reduced thickness leads to discontinuous film that is not conductive to allow for the CV and ETS measurements. In parallel, from the scan-rate dependence measurements, we also observed in the new experiments that the influence of scan rates was gradually reduced with the decreasing scan rates (due to the longer reaction times that allow for more sufficient reaction toward equilibrium states), as clearly seen in new Figure S21e-h. We can therefore conclude that 10 mV/s is a scan rate value that is low enough to minimize its influence on diffusion-controlled shifts in obtained electrochemical potentials. Scan-rates smaller than 10 mV/s led to the instability of the device, especially for the minimum thickness of ~ 25 nm. In summary, while we are aware of the possible impact of electrolyte diffusion on the absolute value of H_{abs} potentials, the potential values obtained from 25 nm thickness device and with 10 mV/s scan rate represent the closest value to the intrinsic properties of the electrode materials, where the impact from the insufficient electrolyte diffusion to the sub-layer

nanowires in the thin film device was minimized, and the measured onset potential value for phase transition was solid.

With the data shown in **new Figure S18 and new Figure S21**, we have further summarized the phase transition potentials to better reveal the intrinsic properties of catalysts and electrolytes with precisely controlled experimental factors. The data was added as **new Figure 3n** in the revised manuscript. It can be clearly seen that, even with the standard deviation from different scan rates, the statistically average phase transition potential of Pd was about 150 mV more positive than that of Pd₄Ag in KHCO₃, reflecting the different M–H interactions. Additionally, by switching the electrolyte from KHCO₃ to K₂HPO₄/KH₂PO₄, the phase transition potentials of both Pd and Pd₄Ag markedly shift to 0~0.1 V_{RHE}, reflecting the strong proton-donation and fast H sorption kinetics in K₂HPO₄/KH₂PO₄. This summary indicates that the comparison between different materials and electrolytes were scientifically solid when the experimental conditions were kept constant (film thickness, potential scan rates, etc.), and further reveal that the conclusion to distinguish the intrinsic properties of catalysts and electrolytes becomes more convincing using systematic and statistical evaluations. Again, we thank the reviewer for the valuable suggestions.

The new data have been updated as **new Figure 3n, S18 and S21** in the revised manuscript, and the corresponding discussions have been updated in the manuscript, as the following:

Page 12, line 8: “ETS measurements were further conducted to elucidate the H sorption kinetics and rationalize the distinct CO₂RR performance of different Pd-based catalysts in different electrolyte environments. First, we have compared the H sorption kinetics by ETS in Ar-saturated KHCO₃ and HClO₄ at varying scan rates (Figure 3d & S18). In HClO₄, significantly larger ETS hysteresis loops were observed to achieve complete H adsorption/desorption when the scan rate was increased from 10 to 80 mV/s, which confirms that the H sorption process is kinetic-dependent. In addition, while the degree of phase transition (corresponding to the ETS current level) kept unchanged in HClO₄, lower degree of phase transition (dashed arrow in Figure 3d) was observed in KHCO₃ with the increasing scan rate. These results suggest the slow H sorption kinetics in near neutral electrolyte, which is reasonable due to the low concentration of hydronium ion and slow kinetics of water reduction in neutral/alkaline electrolytes⁴⁶⁻⁴⁹.”

Page 16, line 17: “As summarized in Figure 3m, while the H sorption kinetics is inhibited by CO₂RR-related H consumption and/or CO poisoning, it can be promoted by the local proton-donating species including H₂CO₃ in equilibrium with CO₂. It should also be noted that the H sorption is essentially a kinetic-dependent process, and the phase transition potentials may vary under different test conditions (potential scan rates, electrode geometries, etc.). Additional ETS tests indeed show positive shifts of the phase transition potential at smaller film thickness or slower scan rates (Figure S21), emphasizing the importance of consistence in test conditions. To this end, the ETS measurements conducted with 25 nm thin film thickness and 10 mV/s scan rate represent the experimental condition that gives close-to-intrinsic properties of the electrode materials, where the impact from the insufficient electrolyte diffusion to the sub-layer nanowires within the thin film device was minimized. With precise control of these experimental factors, the phase transition potential of Pd was determined to be about 150 mV more positive than that of Pd₄Ag in KHCO₃, reflecting the different M–H interactions. Additionally, by switching the electrolyte from KHCO₃ to K₂HPO₄/KH₂PO₄, the phase transition potentials of both Pd and Pd₄Ag

markedly shift to $0 \sim 0.1 V_{\text{RHE}}$, reflecting the strong proton-donation and fast H sorption kinetics in $\text{K}_2\text{HPO}_4/\text{KH}_2\text{PO}_4$. The key results and the corresponding conclusions were summarized in Figure 3n.”

New Figure S18. $I_{\text{SD}}-V_G$ (ETS) curves of Pd in 0.1 M Ar-saturated HClO_4 (a) and KHCO_3 (b) with different scan rates at 10, 20, 40 and 80 mV/s. Film thickness is ~ 200 nm. Dashed arrows indicate the influence of scan rate (i.e., dynamic reaction time) on the hysteresis and degree of Pd phase transition. Solid arrows in all figures indicate the potential sweeping direction.

New Figure S21. (a-d) Phase transition responses of Pd (a, b) and Pd_4Ag (c, d) in 0.1 M CO_2 -saturated KHCO_3 (a, c) and $\text{K}_2\text{HPO}_4/\text{KH}_2\text{PO}_4$ (b, d) with different scan rates and film thicknesses. (e-f) Summarized dependence of onset

potential for the phase transition of Pd (e, f) and Pd₄Ag (g, h) on the potential scan rate with different film thicknesses.

New Figure 3n. Summary of phase transition potentials of Pd (blue) and Pd₄Ag (red) under CO₂RR conditions in KHCO₃ and K₂HPO₄/KH₂PO₄ obtained at 10 mV/s with different film thicknesses.

(2) The authors make several claims that the electronic structure of the Pd₄Ag alloy is responsible for both the reduced CO poisoning and the reduced H absorption capacity. Based on their data, however, a simple geometric effect can explain their observations. The stoichiometry of Pd₄Ag₁ implies that, in a homogeneous alloy, every palladium atom in an fcc lattice is in contact with a Ag atom. Ag is known to both rapidly evolve CO (due to the low CO binding energy) and does not absorb hydrogen into its lattice. A silver atom near every Pd may therefore inhibit CO poisoning and H absorption-based only on its proximity and not on orbital hybridization. If the authors are going to claim electronic structure is the origin of their observation, they need to supplement their data with additional computational evidence showing such an effect.

Reply: We thank the reviewer for this insightful input and valuable suggestion. To further reveal the origin of the reduced CO poisoning and H absorption capacity on Pd₄Ag, we have conducted the density-functional theory (DFT) calculations as suggested by reviewer, which lead to better understanding of the system.

Specifically, DFT calculations were conducted with the implicit solvent model to study the binding energy of *H and *CO. The supercell of (√13×√7) R19° with five atomic layers was chosen to construct the Pd (111) surface. Two Pd atoms were replaced by Ag atoms in each atomic layer to construct the Pd₄Ag surface (new Figure S13). To calculate the binding energies, the H atom and vertical CO molecule were placed at a distance of 1.5 Å and 1.8 Å from the surface. Adsorption energies of *H and *CO on Pd₄Ag and Pd surfaces at different sites were further obtained.

As shown in new Figure S13a, c and new Table S1, it is obvious that the *H adsorption on Pd₄Ag at the Pd top side (N.A., such adsorption configuration is unstable due to the high energy level) is weakened as compared to the same top site on pure Pd surface (adsorption energy 0.16 eV), demonstrating the impact from the changing

electronic structure of Pd atoms in Pd₄Ag. Similarly, the *H adsorption energy on the Pd₃ hollow side (with no engaging Ag atom) are slightly more positive (-0.19/-0.17 eV) than that on pure Pd surface (-0.20/-0.17 eV), indicating the greatly reduced long-range impact of changing electronic structure of Pd atoms in Pd₄Ag. These adsorption energy characteristics on the Pd adsorption sites that do not involve the Ag atoms serve as evidence to support the presence of the charge transfer between Ag and Pd in the alloy lattice, which can be further confirmed by the pDOS calculations (shown in new Figure S13g). The charge transfer from Ag to Pd can also be supported by similar calculations in recent reports (*Adv. Mater.* **2020**, *32*, 2000992; *J. Am. Chem. Soc.* **2019**, *141*, 4791–4794). To further probe the possible proximity effect as suggested by the reviewer, we continue to analyze the *H binding at the Ag-involved hollow sites on Pd₄Ag. A more obviously weakened adsorption energy (-0.10 eV) was observed, indicating that indeed a Ag atom near Pd can inhibit the H adsorption (and thus absorption) simply based on its proximity (i.e., a geometric effect). The analysis of *CO binding leads to similar conclusion. As also shown in the new Figure S13b, d and new Table S1, the binding energy of *CO on the top site of Pd atom adjacent to a Ag atom in Pd₄Ag (-0.55 eV) is more positive compared to top site on pure Pd (-0.62 eV), while the *CO binding energy at Pd₃ hollow sites are similar on the two surfaces, indicating that the alternation in Pd electronic structure can reduce the CO poisoning on Pd₄Ag, yet the impact is significantly reduced from the long-range perspective. Additionally, CO further show a considerably reduced binding energy at the Pd₂Ag hollow sites (N.A., such adsorption configuration is unstable due to the high energy level), as compared to the *CO binding energy at hollow sites on pure Pd (-1.09/-1.11 eV), indicating that the Ag atom near Pd can inhibit CO poisoning based on both charge transfer and geometric effects. It should also be noted that, at the three-atom hollow site that involves both Ag and Pd atoms, the charge transfer between Ag and the adjacent Pd is inevitable, and therefore the impacts of electronic structure change and geometric effects presumably co-exist, which is also consistent with our previous statements (*Adv. Mater.* **2021**, *33*, 2005821). Based on the quantitative comparison of binding energies in different configuration, we can further conclude that at least at the hollow site adsorption configuration, the impact of the proximity effect is more dominant than the change of electronic structure. Overall, we thank the reviewer for making such valuable suggestion that promotes our understanding on this issue. In this study, our ETS results have shown a reduced *in situ* phase transition degree and CO poisoning effect in Pd₄Ag during electrochemical processes as compared to pure Pd, experimentally confirming the weakened M-H and M-CO interactions in Pd₄Ag. The corresponding calculation results were added as new Figure S13 and new Table S1 in the revised SI, and the corresponding discussions were added in the revised manuscript, as the following:

Page 9, line 9: “As for Pd₄Ag alloying catalyst, the onset potential for H absorption is at 0.074 V_{RHE}, which is close to that of Pd. However, the *I*_{SD} drop in response to hydride formation is considerably lower (Figure 2c), and the $\Delta R_{(Pd_4Ag)H_x}$ is calculated to be only 0.14%, which is about 4.12% of ΔR_{PdH_x} , clearly indicating a significantly different M-H interaction compared to pure Pd⁴⁴. As shown in Figure 2d, the $\Delta R_{(Pd_4Ag)H_x}$ is lower than that of pure Pd at each film thickness, with a theoretical intercept of 13%. The M-H interactions in Pd₄Ag and pure Pd were further revealed by DFT calculations, as shown in Figure S13 and Table S1. With a much weaker Ag-H interaction, the alloying Ag atoms can reduce the H adsorption and absorption in Pd₄Ag from both electronic structure and proximity effects. Based on the on-chip ΔR_{MH_x} of Pd and Pd₄Ag, we can further estimate the number of the absorbed H atoms by referring to the known relationship between resistivity and H/M ratio (H/M represents the ratio of hydrogen atoms to the combined total of Pd and Ag atoms) of Pd₈₀Ag₄₀ alloy and pure Pd (Figure S14)^{43, 45}.

The on-chip ΔR_{PdH_x} of 45% corresponds to H/M ratio of 0.42, and $\Delta R_{(\text{Pd}_4\text{Ag})\text{H}_x}$ of 13% corresponds to H/M ratio of 0.29. The lower *in situ* H/M ratio of Pd₄Ag shows that the doping of Ag weakens the phase transition from Pd to Pd hydride, which provides a solid experimental evidence to the weakened M–H interaction and thus phase transition.”

Page 15, line 1: “Similarly, Figure 3f depicts the CV and ETS curves of Pd₄Ag in 0.1 M Ar- and CO₂-saturated KHCO₃ that reveal the H sorption kinetics during CO₂RR. The onset potential for phase transition shifted negatively from $-0.145 V_{\text{RHE}}$ (i in Figure 3h) to $-0.324 V_{\text{RHE}}$ (ii in Figure 3h) after the introduction of CO₂ in the electrolyte, and both are considerably lower than that of pure Pd. The more negative phase transition potentials of Pd₄Ag further prove its weakened M–H interaction after Ag alloying. Additionally, the CO poisoning effect during CO₂RR is not observed on ETS, and H sorption kinetics is largely hindered due to its consumption for intense formate production (dashed arrow in Figure 3h). These *in situ* observations provide convincing evidence for the strong CO poisoning resistance in Pd₄Ag alloying catalysts, which results in the considerably improved conversion rate and stability for CO₂RR (Figure S15&S16). As shown in Figure S13&Table S1, our DFT calculation results also confirm that Ag can reduce the binding energy of poisonous *CO at its surrounding sites. Finally, for similar reason to the Pd case, the inhibition by CO₂RR leads to a smaller ETS hysteresis loop for H sorption in Pd₄Ag (indicated by dashed arrow in Figure 3f).”

Page 17, line 24: “Interestingly, similar trend was not observed in Pd₄Ag, demonstrating the unique H sorption thermodynamics in response to the H⁺ concentration in this alloying structure (blue bars in Figure 4c). First, due to the alternation in the electronic structure of Pd and the proximity effect after Ag doping, the H binding energy is reduced on Pd₄Ag surface, leading to the lower H/M ratio in Pd₄Ag as compared to pure Pd (Figure 4c).”

In addition to the Ag doping effect, the H atom in sub-surface hydride layer can serve as a doping element that significantly increase the electron density of Pd and reduces the CO adsorption energy (*J. Am. Chem. Soc.* **2018**, *140*, 2880–2889; *Adv. Energy Mater.* **2019**, *9*, 1802840). In this study, Pd₄Ag was found to undergo obvious phase transition (H/M: 0.36~0.43) during CO₂RR, contributing to its high current density and stability for formate production. Besides, for Pd₄Ag that intrinsically has weak M–H interaction, H sorption kinetics and phase transition can be further promoted by utilizing electrolytes with high proton-donating capacity, and finally promote CO₂RR. The correlated discussions have been updated in the revised manuscript, as the following:

Page 21, line 22: “**Connection between CO₂RR performances and H sorption processes.** The dynamic, equilibrium and transient quantification of subsurface H/M level (and therefore the H sorption kinetics) can be correlated to the CO₂RR performance under each specific condition. We first performed the density functional theory (DFT) calculations (Figure S14) to rationalize the reduced CO poisoning and high formate production activity on Pd₄Ag surface (as compared to pure Pd), demonstrating the role of charge transfer and geometric effects on the reduced M–H interaction and CO poisoning²⁰. In addition, we found that both Pd and Pd₄Ag exhibit different CO₂RR performances in varying H-donating electrolytes (Figure 3a-c). K₂HPO₄/KH₂PO₄ can accelerate the H sorption kinetics, which is reflected by the increased phase transition degree (H/M ratio) in Pd₄Ag (from 0.38 in KHCO₃ to 0.48 in KH₂PO₄/K₂HPO₄) during CO₂RR (see Figure 4c and Table S2). Correspondingly, an increase in formate FE of about 8% on Pd₄Ag was obtained at $-0.4\sim-0.2 V_{\text{RHE}}$ (Figure 3a), proving that strong proton supply in

$\text{KH}_2\text{PO}_4/\text{K}_2\text{HPO}_4$ electrolyte and high degree in operando phase transition (H doping) would be beneficial for reducing CO poisoning and promoting formate production^{18, 24, 28} (Figure 3b). As a result, Pd_4Ag demonstrated a significantly enhanced current density and stability for formate production compared with pure Pd (Figure 3a, b). It should also be mentioned that the enhanced proton reduction kinetics in $\text{KH}_2\text{PO}_4/\text{K}_2\text{HPO}_4$, as revealed by ETS investigation, simultaneously promotes the competing HER (Figure 3c)^{50, 52} and thus leads to obviously diminished formate FE (Figure 3a) at low overpotential region ($-0.15 \sim 0 \text{ V}_{\text{RHE}}$), where CO poisoning and CO_2RR is relatively weak and less dominant. As a comparison, the H/M ratios of pure Pd in the CO_2 -saturated $\text{KH}_2\text{PO}_4/\text{K}_2\text{HPO}_4$ and KHCO_3 (0.48 and 0.51) are close due to its originally strong Pd-H interaction, and $\text{KH}_2\text{PO}_4/\text{K}_2\text{HPO}_4$ mostly accelerates the HER activity and the formate FE are thus reduced (Figure 3a). Therefore, both CO_2RR and HER performances can be modulated by electrolyte environments that fundamentally determines the H sorption equilibrium and kinetics, and it is important to strike a balance among different and comprehensively connected near-surface, on-surface and sub-surface reactions, as summarized in Figure 4e.”

New Figure S13. (a-d) Adsorption configurations of $^*\text{H}$ (a, c) and $^*\text{CO}$ (b, d) on Pd_4Ag (a, b) and pure Pd (c, d). The

adsorption energies at different sites are listed in the red cycles. The unit of adsorption energy is “eV”. “N/A” in yellow cycles represent unstable adsorptions and their adsorption energies cannot be obtained during DFT calculations. Pd atoms are blue and Ag atoms are grey. (e-f) Adsorption energies of *H and *CO on Pd₄Ag and Pd surfaces at different sites. (g) Partial density of states (PDOS) of Pd₄Ag and Pd.

New Table S1. Adsorption energies of *H and *CO on Pd₄Ag and Pd surfaces at different sites. “N/A” represents unstable adsorptions and their adsorption energies cannot be obtained during DFT calculations.

Adsorbate	*H (top)	*H (hollow) (@Ag)	*H (hollow) (@Pd)
Pd	+0.16 eV	---	-0.17/-0.2 eV
Pd ₄ Ag	N/A	-0.1 eV	-0.17/-0.19 eV
Adsorbate	*CO(top)	*CO (hollow) (@Ag)	*CO (hollow) (@Pd)
Pd	-0.62 eV	---	-1.09/-1.11 eV
Pd ₄ Ag	-0.61/-0.55 eV	N/A	-1.1/-1.12 eV

Page 26, line 7: **Computational Method.** The DFT calculations were performed using the revised Perdew-Burke-Ernzerhof functionals (RPBE) (*Phys. Rev. B* **1999**, *59*, 7413–7421) of generalized gradient approximation (GGA) implemented in the Vienna Ab-initio Simulation Package (VASP) code (*Phys. Rev. Mater.* **2019**, *3*, 044001; *Phys. Rev. B* **1996**, *54*, 11169). The projector-augmented wave (PAW) method (*J. Phys. Condens. Matter* **1994**, *6*, 8245; *Phys. Rev. B* **1994**, *49*, 16223) was applied to describe the electron-ion interactions. A kinetic energy cutoff for the plane wave expansions was set to be 520 eV. The method of Methfessel-Paxton (MP) was applied and the width of the smearing was chosen as 0.2 eV. The supercell of (√13×√7) R19° with five atomic layers was chosen to construct the Pd (111) surface, and two Pd atoms were replaced by Ag atoms in each atomic layer to construct the Pd₄Ag surface. More than 10 Å of vacuum space was used to avoid the interaction of the adjacent images. For sampling the reciprocal space, k-points of Γ-centered 4×3×1 were used for surface calculations. All structures were fully relaxed until the force components were less than 0.03 eV·Å⁻¹. Implicit solvent model was used in our calculations by VASPsol (*J. Chem. Phys.* **2014**, *140*, 084106; *J. Chem. Phys.* **2019**, *151*, 234101). A Debye screening length of 9.61 Å was chosen, which corresponds to a bulk ion concentration of 0.1 M. The non-electrostatic parameter, TAU, was set to zero for the purpose of convergence.

The adsorption energy of CO is defined as:

$$E_{b,CO} = E_{*CO} - E^* - E_{CO}$$

where E_{*CO} , E^* and E_{CO} are the total energy of the surface with adsorbed CO, pristine surface and CO molecule in the gas phase, respectively.

The adsorption energy of H is defined as:

$$E_{b,H} = E^*H - E^* - 1/2E_{H_2}$$

where E^*H , E^* and E_{H_2} are the total energy of the surface with adsorbed H, pristine surface and H_2 molecule in the gas phase, respectively.”

Minor Points:

(1) Figure 2, panel a,b,c. Please clearly indicate which y-axis corresponds to the plotted data. As it currently stands, it is very difficult for the reader to interpret these plots.

Reply: We thank the reviewer for pointing out this issue of clarity. Per reviewer’s suggestion, we have updated the new Figure 2a-c with added axis titles and arrows indicating the correspondence between each plot and the linked y axis in the revised manuscript, as shown below.

New Figure 2. Electrochemical interfacial H processes in acidic condition. a-c $I_{SD}-V_G$ (ETS, solid) and I_G-V_G (CV, dash) curves of Pt (a), Pd (b) and Pd₄Ag (c) in 0.1 M HClO₄. Film thickness is ~200 nm. I, II and III represent the different states of the metals. Solid arrows in a-c indicate the potential sweeping directions.

(2) Figure S15 panels a and c. The arrows seem to be going in opposite directions with increasing scan. Please alter the figure to clearly indicate the scan direction.

Reply: We thank the reviewer for pointing out this issue. Per reviewer’s suggestion, we have added positive and negative arrows for the first and final cycle of the successive ETS curves in new Figure S6 and S19 in the revised SI, as shown below.

New Figure S6. I_{SD} - V_G (ETS) and I_G - V_G (CV) curves during electrochemical activation process (each sweep contains two potential cycles) of Pt (a, b), Pd (c, d) and Pd₄Ag (e, f) in 0.1 M HClO₄ before standard CV and ETS measurements. Solid arrows in all figures indicate the potential sweeping directions.

New Figure S19. I_{SD} - V_G (ETS) and I_G - V_G (CV) curves (each sweep contains two potential cycles) of Pd (a, b) and Pd₄Ag (c, d) in 0.1 M CO₂-saturated KHCO₃. Solid arrows in all figures indicate the potential sweeping directions.

(3) In the text and Figure 4a, the authors switch notation between the empirical chemical formula and an abbreviated name. Specifically, H₂CO₃ and 'PBS'. To improve the readability of the figures and text, the authors

should choose either the chemical formula or abbreviations, but not both.

Reply: We thank the reviewer for pointing out this issue. To improve the readability of the figures and text, we have unified the format of the electrolytes as full name and empirical chemical formula: perchloric acid (HClO_4), bicarbonate buffer (KHCO_3), phosphate buffer ($\text{K}_2\text{HPO}_4/\text{KH}_2\text{PO}_4$). All the corrections have been highlighted in the revised manuscript.

REVIEWER COMMENTS

Reviewer #1 (Remarks to the Author):

The authors have addressed the majority of my concerns.

I believe the paper is now fit for publication. However I would request that the authors address one final point.

In the discussion of Figure 3, the authors discuss whether CO poisoning may be occurring on each catalyst (Pd and Pd₄Ag) during CO₂ reduction; this discussion is unclear.

My understanding is that with Pd, the authors observe either a shift in the potential of or a decrease in the amount of H₂ adsorption in Pd, and argue this could be due to CO poisoning (preventing H₂ adsorption and therefore H₂ adsorption). With Pd₄Ag, it sounds like they see a similar shift/decrease, but attribute it instead to the rapid consumption of H₂ adsorption to form formate, instead of CO poisoning. The authors go so far as to say that this "provides convincing evidence" for the strong CO poisoning resistance of Pd₄Ag.

First, it is not really clear to me how the authors are interpreting the ETS measurements to see CO poisoning with Pd and a lack of CO poisoning with Pd₄Ag.

Second, if the effect of switching from Ar to CO₂ does have a similar effect on the H₂ adsorption hysteresis loop for both Pd and Pd₄Ag, then I don't think this result could be used to conclude CO poisoning is happening with one catalyst and not with the other.

Please clarify and/or correct.

Reviewer #2 (Remarks to the Author):

The revisions provided by the authors have addressed the my original concerns with this manuscript. I believe it is now suitable for publication.

Reviewer #1:

The authors have addressed the majority of my concerns. I believe the paper is now fit for publication. However I would request that the authors address one final point.

In the discussion of Figure 3, the authors discuss whether CO poisoning may be occurring on each catalyst (Pd and Pd₄Ag) during CO₂ reduction; this discussion is unclear. My understanding is that with Pd, the authors observe either a shift in the potential of or a decrease in the amount of H absorption in Pd, and argue this could be due to CO poisoning (preventing H_{ads} and therefore H_{abs}). With Pd₄Ag, it sounds like they see a similar shift/decrease, but attribute it instead to the rapid consumption of H_{ads} to form formate, instead of CO poisoning. The authors go so far as to say that this “provides convincing evidence” for the strong CO poisoning resistance of Pd₄Ag.

First, it is not really clear to me how the authors are interpreting the ETS measurements to see CO poisoning with Pd and a lack of CO poisoning with Pd₄Ag.

Second, if the effect of switching from Ar to CO₂ does have a similar effect on the H_{abs} hysteresis loop for both Pd and Pd₄Ag, then I don't think this result could be used to conclude CO poisoning is happening with one catalyst and not with the other.

Please clarify and/or correct.

Reply: We thank the reviewer for pointing out this issue of clarity in the interpretation of our ETS results. In Figure 3, the effect of switching from Ar to CO₂ indeed have “**similar effects on the H_{abs} hysteresis loops for both Pd and Pd₄Ag**”, emphasized by the shift of phase transition potential and decrease of ETS current (I_{SD}), leading to the alternated ETS loop characteristics. Hence, it is concluded that CO₂RR inhibits H sorption process for both Pd and Pd₄Ag. On the other hand, however, **the analysis that leads to the discussion of CO poisoning actually comes from the more detailed comparison of fine ETS characteristics within the potential range of -0.2 to -0.4 V_{RHE} region.**

First, as shown in the **Figure 3g and 3k**, an expected smooth linear decrease of I_{SD} (black curves, starting from “i” in **Figure 3g&3k**) can be observed on Pd in Ar-electrolyte due to the gradually increased level of H_{abs}, which serves as a normal baseline of the H sorption kinetics in Pd. In sharp contrast, we found that this smooth behavior changes to a distinctive nonlinear behavior during CO₂RR, with turning points emphasized by “ii, iii and iv” on red curves in **Figure 3g&3k**. Based on the product analysis in different potential range and existing reaction mechanism of Pd-catalyzed CO₂RR, we further assign the different fine regions as the following. For pure Pd: (i) is the onset of phase transition in Ar; (ii) is the onset of phase transition in CO₂, where the (i→ii, iii) shift is resulted from H consumption along with CO₂RR, and after (iii) the increased H kinetics overweigh the H consumption from CO₂RR to give another smooth declining region (iii→iv); and **(iv) H sorption during CO₂RR accompanied by the presumable surface CO poisoning at high overpotential, where the emerging CO_{ads} reduces the H kinetics and hence the declining slope to form a turning point iv.** In short, we propose here the appearance of the turning point iv and the nonlinear ETS behavior during CO₂RR in this fine region is indicative of CO poisoning.

Interestingly for Pd₄Ag, the variation trend of fine ETS in the same potential region (**Figure 3h and 3l**) is relatively simple, and no obvious switch of ETS behavior corresponding to the CO poisoning (emergence of turning point iv and the nonlinear ETS behavior) is observed. Hence, we concluded that there is “**a lack of CO poisoning with Pd₄Ag**” based on the ETS characteristics of Pd/ Pd₄Ag under CO₂RR conditions, which is consistent with its high working stability and the DFT calculation results, both reported in this work or in literatures.

Overall, ETS results show the competition between H sorption and different surface reactions (CO₂RR and CO

poisoning). While we are confident with the precision and reproducibility of these fine ETS characteristics, we would also agree with reviewer that the interpretation of the ETS measurements and surface competition processes is not fully “**evidencing**”, and it is uncertain whether there is absolutely no CO poisoning on Pd₄Ag only based on ETS results. Meanwhile, this is also not the major point/conclusion in the work. Further rigorous proof of the *in situ* CO_{ads} may require the use of other *in situ* spectroscopies (such as FTIR) that match micro/nano device platform under CO₂RR conditions, and the concurrent ETS measurement can then provide strong support from another perspective and help to quantify this process. This is certainly the future direction we will pursue. At this point, to better address this issue and to prevent the over-interpretation of the ETS data, we have now revised the discussion of Figure 3 to make the description more clarified and rigorous. The **new Figure 3g,h,k,l** were slightly modified for easier visualization, and new discussions have been updated in the manuscript, as the following:

Page 14: “Importantly, these H-involving processes during CO₂RR can be reflected on the ETS signals regarding H sorptions. On this basis, as the unusual change in I_{SD} (Figure 3g) is in well correspondence with the high formate FE (>70%) in the potential range $>-0.2 V_{RHE}$, our results therefore confirm that proton consumption and site blocking by intermediate adsorptions during formate production significantly reduce the H diffusion kinetics and level of phase transition under scanning potential (non-equilibrium) condition. When the potential continues to decrease to **a more negative potential ($-0.3 V_{RHE}$, indicated by iv in Figure 3g), the decline of I_{SD} is slowed down (leading to a clear two-stage, non-linear ETS characteristic within the range of -0.2 to $-0.4 V_{RHE}$)** probably due to severe CO poison, which inhibits the production of H_{ads} and subsequent H absorption³².”

Page 15: “Similarly, Figure 3f depicts the CV and ETS curves of Pd₄Ag in 0.1 M Ar- and CO₂-saturated KHCO₃ that reveal the H sorption kinetics during CO₂RR. The onset potential for phase transition shifted negatively from $-0.145 V_{RHE}$ (i in Figure 3h) to $-0.324 V_{RHE}$ (ii in Figure 3h) after the introduction of CO₂ in the electrolyte, and both are considerably lower than that of pure Pd. The more negative phase transition potentials of Pd₄Ag further confirm its weakened M–H interaction after Ag alloying. Additionally, **no change of fine ETS characteristics at high overpotentials (-0.2 to $-0.4 V_{RHE}$) was observed, which indicates an obvious CO poisoning effect**, and H sorption kinetics **is expectedly** hindered due to **the H** consumption for intense formate production (dashed arrow in Figure 3h). These *in situ* observations **are also consistent with** the strong CO poisoning resistance in Pd₄Ag alloying catalysts, which results in the considerably improved conversion rate and stability for CO₂RR (Figure S15&16). As shown in Figure S13 and Table S1, our DFT calculation results also confirm that Ag can reduce the binding energy of poisonous *CO at its surrounding sites^{20-21, 25}. Finally, for similar reason to the Pd case, the inhibition by CO₂RR leads to a smaller ETS hysteresis loop for H sorption in Pd₄Ag (indicated by dashed arrow in Figure 3f).”

Page 16: “In addition to the proton-donating effect, the influence of formate production and CO poisoning (as a result of the CO₂RR process) on H sorption in K₂HPO₄/KH₂PO₄ is also presented on ETS curves of Pd (ii, iii and iv in Figure 3k respectively). **Again for Pd₄Ag, no obvious CO poisoning signal (lack of point iv in Figure 3l) is reflected on ETS in K₂HPO₄/KH₂PO₄.**”

Figure 3. CO₂RR and H sorption processes in KHCO₃ and K₂HPO₄/KH₂PO₄ electrolytes. a-c Potential-dependent selectivity for formate production (a), and current densities for formate production (b) and hydrogen evolution (c) with Pd (red) and Pd₄Ag (blue) catalysts in 0.1 M CO₂-saturated KHCO₃ (solid curves) and K₂HPO₄/KH₂PO₄ (dash curves). d Phase transition responses of Pd in 0.1 M Ar-saturated KHCO₃ and HClO₄ with varying scan rates. Film thickness is ~200 nm. e-h On-chip I_{SD} - V_G (ETS, solid) and I_G - V_G (CV, dash) curves of Pd (e, g) and Pd₄Ag (f, h) in 0.1 M Ar- and CO₂-saturated KHCO₃. g and h depict the enlarged negative-potential-sweeping ETS curves (0.2 to -0.5 V_{RHE}) in e and f, respectively. i-l On-chip I_{SD} - V_G (ETS, solid) and I_G - V_G (CV, dash) curves of Pd (i, k) and Pd₄Ag (j, l) in 0.1 M Ar- and CO₂-saturated K₂HPO₄/KH₂PO₄. k and l depict the enlarged negative-potential-sweeping ETS curves (0.2 to -0.5 V_{RHE}) in i and j, respectively. Solid arrows in d-l indicate the potential sweeping direction. Film thickness is ~200 nm. m Schematic illustration of different M-H states in KHCO₃ (left panel) and K₂HPO₄/KH₂PO₄ (right panel), and the corresponding CO₂RR processes at the interfaces. n Summary of phase transition potentials of Pd and Pd₄Ag under CO₂RR conditions in KHCO₃ and K₂HPO₄/KH₂PO₄ with different scan rates at 10, 20, 40 and 80 mV/s. Film thickness is ~25 nm.